# Non-cell-autonomous regulation of germline proteostasis by insulin/IGF-1 signaling-induced dietary peptide uptake via PEPT-1

Tahir Muhammad [1], Stacey L Edwards[2], Allison C Morphis [2], Mary V Johnson[1], Vitor De Oliveira [1], Tomasz Chamera[2], Siyan Liu[1], Ngoc Gia Tuong Nguyen[1] & Jian Li [1✉]

## Abstract

**Gametogenesis involves active protein synthesis and is proposed to rely on proteostasis. Our previous work in *C. elegans* indicates that germline development requires coordinated activities of insulin/IGF-1 signaling (IIS) and HSF-1, the central regulator of the heat shock response. However, the downstream mechanisms were not identified. Here, we show that depletion of HSF-1 from germ cells impairs chaperone gene expression, causing protein degradation and aggregation and, consequently, reduced fecundity and gamete quality. Conversely, reduced IIS confers germ cell resilience to HSF-1 depletion-induced protein folding defects and various proteotoxic stresses. Surprisingly, this effect was not mediated by an enhanced stress response, which underlies longevity in low IIS conditions, but by reduced ribosome biogenesis and translation rate. We found that IIS activates the expression of intestinal peptide transporter PEPT-1 by alleviating its repression by FOXO/DAF-16, allowing dietary proteins to be efficiently incorporated into an amino acid pool that fuels germline protein synthesis. Our data suggest this non-cell-autonomous pathway is critical for proteostasis regulation during gametogenesis.**

**Keywords** Germline Proteostasis; Heat Shock Factor 1; Insulin and IGF-1 Signaling; Protein Synthesis; Peptide Uptake
**Subject Categories** Development; Metabolism; Translation & Protein Quality

## Introduction

Gametogenesis, in which primordial germ cells develop into gametes (sperm or oocytes) through mitotic proliferation, meiotic differentiation, and gamete maturation, is essential for sexually reproducing animals to transmit genetic information over generations while introducing genetic diversity (Childs et al, 2008). During gametogenesis, germ cells experience periods of very active protein synthesis to make cell-cycle-specific proteomes through mitosis and meiosis and to prepare maternal proteins for embryogenesis (Muhammad and Li, 2023; Sala and Morimoto, 2022). These newly synthesized proteins must be folded into proper conformations assisted by molecular chaperones to be functional (Labbadia and Morimoto, 2015). How animals coordinate protein translation and folding during gametogenesis to ensure sufficient folding capacity is poorly understood.

Heat shock factor 1 (HSF1), a key transcription regulator of cellular responses to proteotoxic stress (Gomez-Pastor et al, 2018; Pessa et al, 2024), has evolutionarily conserved roles in gametogenesis in vertebrates and invertebrates (Christians et al, 2000; Edwards et al, 2021; Jedlicka et al, 1997). It activates the transcription of selective molecular chaperones and co-chaperones in germ cells under physiological and stress conditions (Bierkamp et al, 2010; Das et al, 2020; Edwards et al, 2021; Metchat et al, 2009). Reports from *C. elegans* and mice also suggest that HSF1 binds to the promoters and regulates the expression of genes with other essential functions in reproduction, including meiosis (Edwards et al, 2021; Le Masson et al, 2011). It is yet to be determined whether the primary role of HSF1 in germline development is to maintain proteostasis.

Our previous study has shown that the requirement for HSF-1, the only HSF in *C. elegans*, in germline development is dictated by the activity of insulin/IGF-1 signaling (IIS) (Edwards et al, 2021). While in the wild-type animals, loss of HSF-1 from the germline through larval development results in sterility, reduced IIS could partially restore fecundity in the absence of HSF-1 in germ cells. This finding implies that reducing IIS may rewire the proteostasis network and render germ cells less dependent on HSF-1-mediated chaperone expression. IIS is a highly conserved nutrient-sensing pathway that promotes gametogenesis both cell-autonomously in the germline and non-cell-autonomously from somatic tissues (Muhammad and Li, 2023; Templeman and Murphy, 2018). In addition, IIS activity needs to be fine-tuned for gamete quality as hyperactivation of IIS has detrimental effects on oogenesis, while reduced IIS preserves functional oocytes during maternal aging in mice and *C. elegans* (Luo et al, 2010; Reddy et al, 2008). On the other hand, IIS impacts proteostasis and responses to proteotoxic stress through transcriptional and post-transcriptional control of

[1]Department of Cell Biology and Anatomy, New York Medical College, Valhalla, NY, USA. [2]Aging and Metabolism Research Program, Oklahoma Medical Research Foundation, Oklahoma City, OK, USA. ✉E-mail: jli37@nymc.edu

many players in protein synthesis, turnover, and quality control (Demontis and Perrimon, 2010; Hsu et al, 2003; Taniguchi et al, 2006; Taylor and Dillin, 2011; Zhao et al, 2007). However, it is not clear why the activities of HSF-1 and IIS need to be coupled for germline development and how IIS regulates germline proteostasis at the organismal level.

In this study, we found that HSF-1 is important for germline proteostasis by enhancing protein folding capacity in *C. elegans* gametogenesis at ambient temperature. Reduced IIS grants germ cells resilience against proteotoxic challenges associated with loss of HSF-1 and stress through lowering ribosomal biogenesis and translation rate. Interestingly, IIS promotes protein synthesis in the germline non-cell-autonomously via transcriptional activation of the intestinal peptide transporter, PEPT-1. Our findings suggest that IIS-mediated dietary protein absorption and germline protein synthesis must work in concert with HSF-1-dependent protein folding to maintain proteostasis in gametogenesis.

## Results

### HSF-1 is important for germline proteostasis at ambient temperature

HSF-1 is well-known for its roles in proteotoxic stress responses, such as against heat shock, but it also regulates gene expression in physiological conditions during development, reproduction, and aging (Edwards et al, 2021; Li et al, 2016; Morphis et al, 2022). In a previous study, we identified the HSF-1 transcriptional program in *C. elegans* germline by combining tissue-specific HSF-1 depletion using the auxin-inducible degron (AID) system with ChIP-seq and RNA-seq analyses at ambient temperature (Edwards et al, 2021). Among those genes that had HSF-1 binding at the promoters and decreased expression upon HSF-1 depletion in the germline of young adults are proteostatic and reproduction-related genes. Kinetic analysis indicated that a group of chaperone and co-chaperone genes were the most sensitive to HSF-1 depletion by showing the quickest decline of mRNA levels (Edwards et al, 2021), implicating that they are likely the primary target genes of HSF-1 in gametogenesis. Interestingly, following the decrease of chaperone mRNAs at 8 h of HSF-1 depletion from the germline, genes in the ubiquitin-proteasome system (UPS) were induced at the 16 h time point (Fig. 1A). This result implies that the UPS is upregulated as a stress response to remove misfolded proteins caused by loss of HSF-1-dependent chaperone expression and decreased protein folding capacity.

To test this hypothesis, we examined the stability of GFP fusion and endogenous proteins followed by HSF-1 depletion (Fig. 1B). Loss of HSF-1 from the germline for 24 h led to a significant decrease of the NMY-2 (non-muscle myosin) fusion protein that is expressed by the NMY-2 promoter and resides in the cytosol of the germline (Fig. 1C,D). This change likely occurred at the protein level as HSF-1 depletion did not lower the mRNA levels of nmy-2 during the 24-h time course (Fig. EV1A). Similarly, the GFP fusion of histone H2B that accumulates in the nuclei of oocytes also decreased upon HSF-1 depletion (Figs. 1E and EV1B). The decline of both NMY-2 and H2B was partially reversed by inhibition of proteasome for 6 h before imaging analysis (Figs. 1C–E and EV1B), suggesting that HSF-1-mediated chaperone expression is essential

for the stability of both cytosolic and nuclear proteins in germ cells. Notably, the short proteasome inhibition was not sufficient to restore HSF-1 protein levels from the depletion by AID (Fig. EV1C). Therefore, the partial recovery of GFP fusion proteins was not due to re-gaining HSF-1 activities in the germline but rather through preventing the degradation of NMY-2 and H2B fusion proteins by the UPS. The endogenous histone H3 showed a similar decline in protein levels upon HSF-1 depletion as measured by immuno-fluorescence (IF) (Fig. 1F,G). In addition, we monitored the change of ubiquitylated proteins in the germline upon loss of HSF-1. Low levels of ubiquitylated proteins were visible in the presence of HSF-1 at the diplotene stage and in oocytes, consistent with the reported functions of protein ubiquitylation in meiosis (Rao et al, 2017; Wu et al, 2021). Upon HSF-1 depletion, levels of ubiquitylated proteins significantly elevated through the germline (Fig. 1F,H), suggesting the increase of misfolded proteins that were ubiquitinylated for clearance. Finally, we found that high molecular weight, ubiquity-lated proteins increased in the detergent-insoluble aggregates upon HSF-1 depletion from the germline, supporting them as the misfolded species and aggregation-prone (Fig. 1I). Interestingly, we also found that a small fraction of α-tubulin was in the insoluble fraction upon HSF-1 depletion, indicating the gain-of-function toxicity of misfolded proteins that trapped other proteins into aggregates and further imbalanced the proteome. The accumulation of ubiquitylated proteins in aggregates also suggests the protein degradation system may be overwhelmed by the continuous influx of misfolded proteins due to insufficient folding capacity. Support-ing this notion, the levels of NMY-2::GFP were recovered after a prolonged 48 h depletion of HSF-1 (Fig. EV1D–F) with the appearance of GFP in puncta-like structures (Fig. EV1E, lower panel).

Our data demonstrate that protein misfolding, degradation, and aggregation increase following the decline of HSF-1-dependent chaperone expression, indicating that HSF-1 is essential for providing protein folding capacity in germ cells.

### Insufficient protein folding capacity compromises fecundity and gamete quality

We then tested the impacts of insufficient protein folding capacity on gametogenesis. As soon as 16 h of HSF-1 depletion from young adult germline, we observed a significant decrease in the portion of mitotic nuclei in the S-phase (EdU positive, Fig. 2A,B). This result is consistent with our previous finding that HSF-1 supports the proliferation of germline stem cells (GSC) (Edwards et al, 2021). In addition, HSF-1 depletion caused an increase of nuclei in the mitotic zone and a decrease in the transition zone (Fig. EV2A,B), suggesting an impairment of GSC differentiation into meiotic prophase I. The defects that we observed in gametogenesis very likely resulted from insufficient protein folding due to the declined expression of HSF-1-dependent chaperone and co-chaperone genes, as at 16 h of HSF-1 depletion, the other HSF-1 target genes just began changing their mRNA levels (Edwards et al, 2021). Loss of HSF-1 from the germline also led to significant oocyte degeneration. During oogenesis, a subset of germ cells at the pachytene stage enter physiological apoptosis, which mainly occurs near the loop region (Fig. 2C, left panel). They are proposed to function as nursing cells and supply cytoplasmic components to the developing oocytes (Gartner et al, 2008). HSF-1 depletion, however, dramatically increased the apoptosis of oocytes

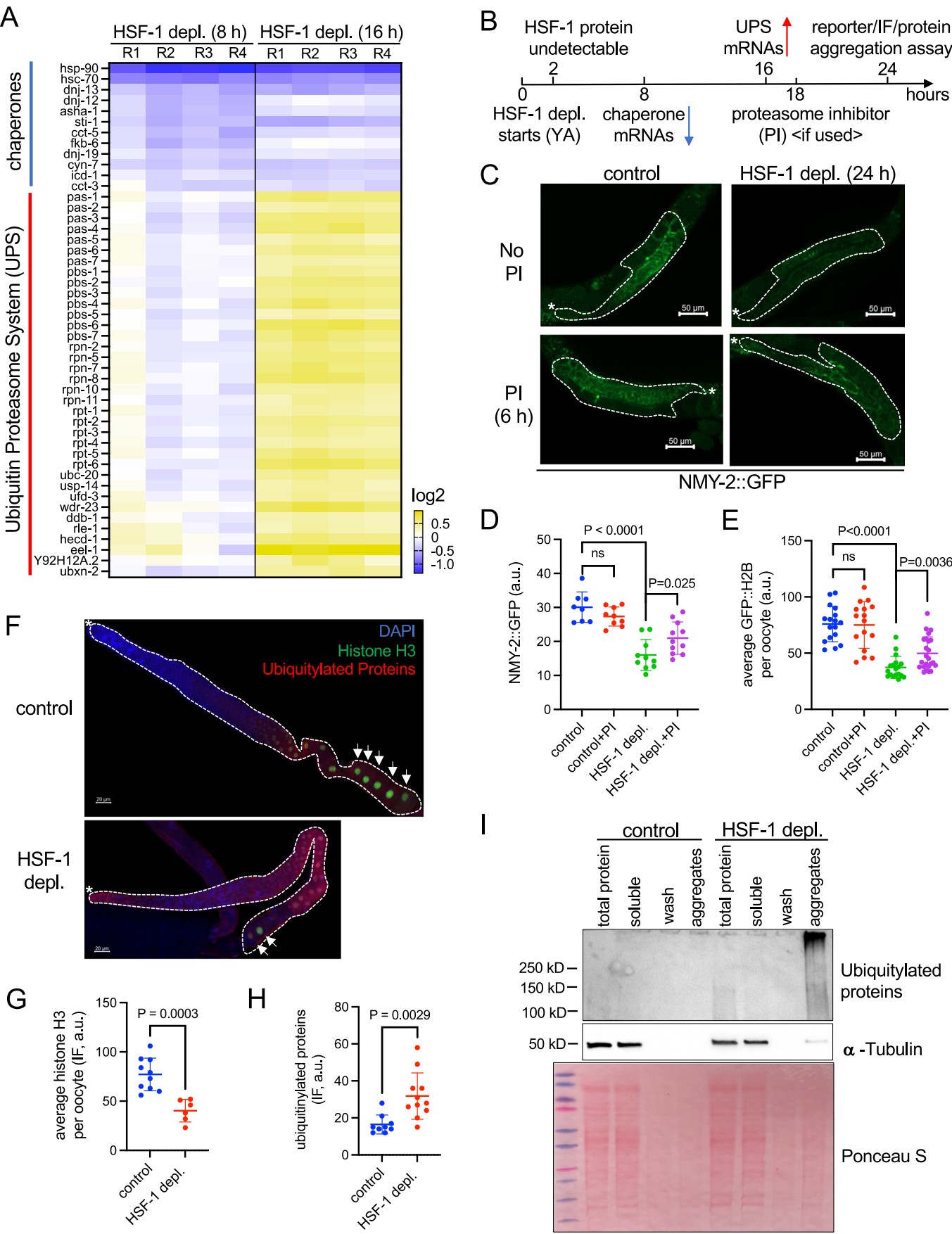

◀  **Figure 1.  Loss of HSF-1 from the germline impairs the expression of chaperone genes, causing protein degradation and aggregation.**

(A) Heatmap of mRNA fold changes at HSF-1-dependent chaperone and co-chaperone genes as well as components of the ubiquitin-proteasome system (UPS) upon depletion of HSF-1 (HSF-1 depl.) from the germline of young adults using auxin-inducible degradation (AID) for 8 h and 16 h. Differentially expressed genes (FDR: 0.05) at either time point in both functional groups are included. The promoters of chaperone and co-chaperone genes shown in the heatmap are bound by HSF-1 in germ cells as determined by ChIP-seq. Heatmap columns represent biological replicates. (B) Schematic diagram of germline-specific HSF-1 depletion experiments that test protein stability, misfolding, and aggregation. The kinetics of HSF-1 depletion and mRNA changes are based on our previous report (Edwards et al, 2021). (C, D) Representative images (C) and quantification of protein levels of NMY-2::GFP transgene (D) in the germline upon HSF-1 depletion. HSF-1 was depleted from the germline of young adults using AID for 24 h in the presence or absence of the proteasome inhibitor bortezomib (PI) for the last 6 h of HSF-1 depletion. The dashed lines outline the gonads. The white asterisks (*) indicate the distal end of gonads where progenitor cells are located. Control: $N \geq 8$; HSF-1 depletion: $N \geq 10$. (E) Quantification of protein levels of GFP::H2B transgene in fully grown oocytes upon germline-specific depletion of HSF-1. The level of GFP::H2B in each oocyte nucleus is quantified, and the average signal of all oocytes for each gonad arm is calculated and plotted. Experiments were done as in (B, C). Control: $N \geq 16$; HSF-1 depletion: $N \geq 18$. (F–H) Representative images (F) and quantification of endogenous histone H3 (G) and ubiquitylated proteins (not free ubiquitin) (H) by immunofluorescence (IF) upon germline-specific depletion of HSF-1 in young adults for 24 h. The dashed lines outline the gonads with the white asterisks (*) marking the distal end. The arrows indicate the fully grown oocytes, where the levels of histone H3 are quantified. For Histone H3, control: $N = 10$; HSF-1 depletion: $N = 6$ (not all gonads examined contained fully grown oocytes upon HSF-1 depletion). For ubiquitylated proteins, control: $N = 9$; HSF-1 depletion: $N = 11$. (I) Western blot of protein aggregates upon germline-specific depletion of HSF-1 in young adults for 24 h. Antibodies against ubiquitin and α-tubulin were used. The 'aggregates' fraction was loaded as 10-fold of the other fractions. In all the dot plots, the mean and standard deviation are plotted. P-values were calculated by unpaired t-test. ns: $p \geq 0.05$. Source data are available online for this figure.

that have finished cellularization and are much bigger in size (Fig. 2C,D). Consistent with the cellular defects, we found the fecundity of self-reproducing hermaphrodites declined dramatically starting from the second day of HSF-1 depletion (Figs. 2E and EV2C). We conclude that it was due to the impairment of ongoing oogenesis in young adults rather than the potential impacts on the quality of sperm that were made at the L4 larval stage because a similar decline of fecundity was observed upon HSF-1 depletion when we mated the hermaphrodites with wild-type males (Figs. 2F and EV2C). Loss of HSF-1 from the germline also compromised the quality of oocytes as embryo lethality increased by more than 10-fold in both self-progenies and mated progenies starting from the second day of HSF-1 depletion (Fig. 2G). Results from germline-specific RNAi against HSF-1-activated chaperone and co-chaperone genes support that the decline in protein folding capacity and loss of proteostasis underlie the reproductive defects upon HSF-1 depletion. Consistent with our previous report (Edwards et al, 2021), knock-down of two essential ATP-dependent chaperones, hsc-70 (also called hsp-1) and hsp-90, from germ cells resulted in sterility or close to sterility (Fig. EV2D–F). RNAi against the co-chaperone gene dnj-13 also significantly reduced brood size and increased embryo lethality, though to a lesser extent than HSF-1 depletion (Fig. EV2D–F). In addition, hsp-90 RNAi increased protein ubiquitylation through the germline and reduced histone H3 protein in oocytes as observed upon HSF-1 depletion (Fig. EV2G–I), suggesting elevated protein misfolding and degradation because of insufficient chaperone levels.

## Reduced insulin/IGF-1 signaling confers germ cells' resilience against limited protein folding capacity

Despite HSF-1's role in germ cell proteostasis, our previous study showed that reduced insulin/IGF-1 signaling (IIS) could alleviate the requirement for HSF-1 in germline development. When HSF-1 was depleted from the germline through larval stages, the wild-type animals were utterly sterile, while the reduction-of-function mutant of insulin/IGF-1 receptor, daf-2(e1370), partially restored reproduction (Edwards et al, 2021). Similar results were observed when we depleted HSF-1 from the germline at the young-adult stage after hermaphrodites switched from spermatogenesis to oogenesis (Fig. 3A,B). Reduced IIS, as in the daf-2(e1370) animals or daf-2(e1370) animals further treated with daf-2 RNAi, decreased the brood size in the presence of HSF-1 (Fig. 3A), consistent with

the role of IIS in promoting gametogenesis (Lopez et al, 2013; Michaelson et al, 2010). However, reduced IIS rendered fecundity less dependent on HSF-1 (Fig. 3A) and dramatically decreased embryo lethality associated with HSF-1 depletion (Fig. 3B). These results implicate that reduced IIS could improve proteostasis in germ cells without HSF-1.

To understand why IIS dictates the requirement for HSF-1 in germ cells, we first tested if reduced IIS restores chaperone expression in the absence of HSF-1 by RNA-seq analysis. The genes encoding hsp-90 and hsc-70 chaperones and two co-chaperones, dnj-13 and sti-1, were still significantly downregulated when HSF-1 was depleted from the germline of daf-2(e1370) animals (Fig. 3C,D). Significantly, upon chronic depletion of HSF-1 starting from egg lay in the daf-2(e1370) animals, the decrease in mRNA levels of those chaperone and co-chaperone genes was comparable with that upon 16 h of HSF-1 depletion in the wild-type animals (Fig. 3C), which was sufficient to cause gametogenesis defects (Fig. 2A,B) and expression changes at ~2000 genes in the wild-type animals (Fig. 3D). Furthermore, RNA-FISH analysis in our previous study has shown although IIS tunes up HSF-1-dependent chaperone expression, in the absence of HSF-1, the wild-type and daf-2(e1370) animals have comparable mRNA levels of hsp-90 and hsc-70 in the germline (Edwards et al, 2021). Therefore, reduced IIS renders germ cells more tolerant to limited chaperone expression.

We then tested if reduced IIS mounts a more robust stress response to loss of HSF-1 from the germline. Reduced IIS enhances stress response through downstream transcription factors, such as FOXO/DAF-16 and NRF/SKN-1 (Henis-Korenblit et al, 2010; Hsu et al, 2003; Murphy et al, 2003; Tullet et al, 2008). Our previous study demonstrates that animals with reduced IIS are tolerant to HSF-1 depletion from the germline in a manner that is independent of SKN-1 and only requires DAF-16 non-cell-autonomously from the soma (Edwards et al, 2021). These data suggest that the stress responses mediated by SKN-1 and DAF-16 are unlikely to play essential roles in the germline to resolve proteotoxicity resulting from the loss of HSF-1. Our RNA-seq analysis showed that the UPS components were not induced upon acute or chronic depletion of HSF-1 from the germline of daf-2(e1370) animals (Fig. 3E). Furthermore, loss of HSF-1 from germ cells only altered the expression of a handful of genes in addition to the four chaperone and co-chaperone genes mentioned above in the daf-2(e1370) animals (Fig. 3D). These results implicate the intrinsic property of

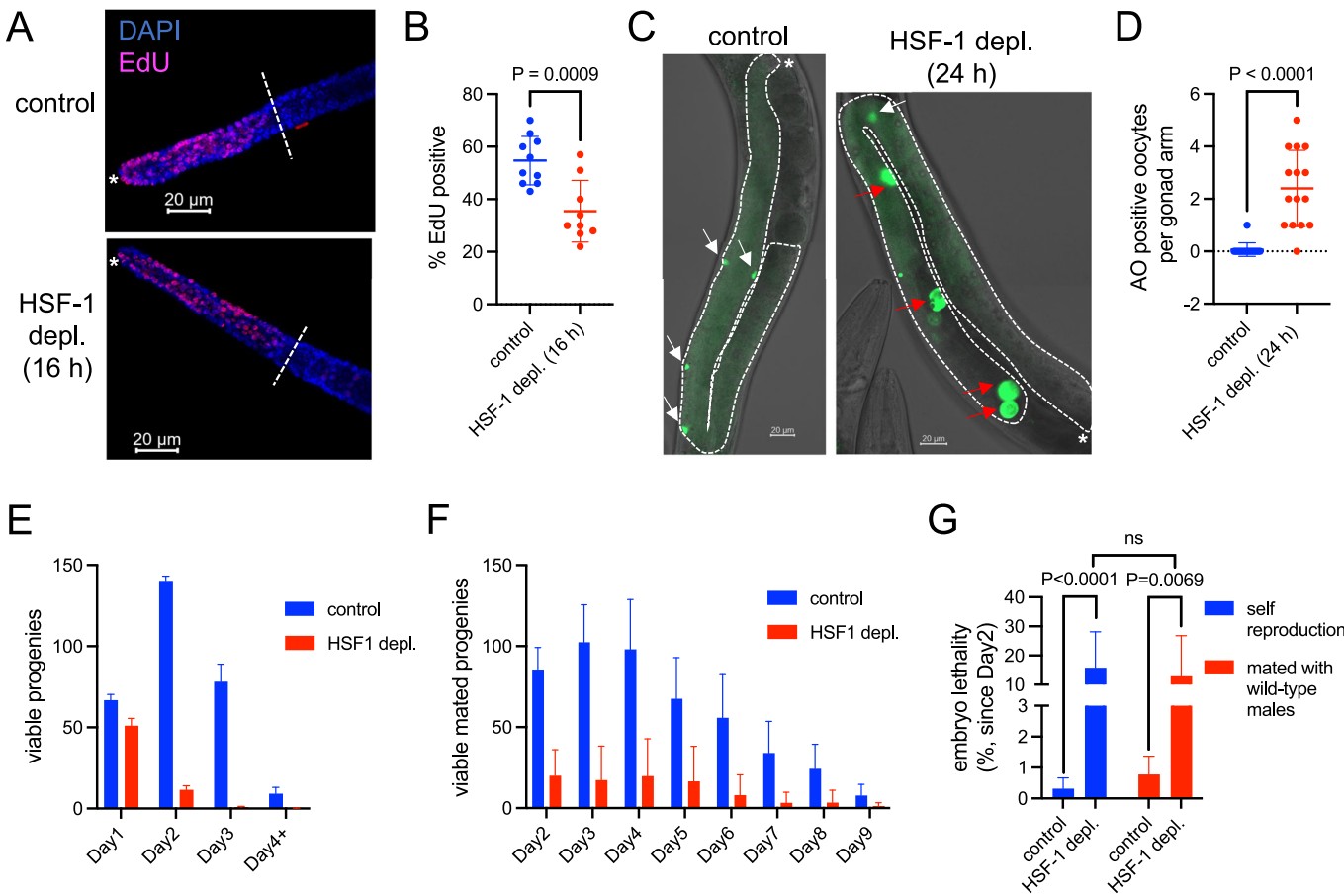

**Figure 2. HSF-1 is required in the adult germline for fecundity and oocyte quality.**

(A) Representative images of gonads with EdU labeling of nuclei in the S-phase and DAPI staining of all germline nuclei upon depletion of HSF-1 from the germline of young adults for 16 h. Progenitor cell proliferation occurs in the mitotic zone between the distal end of gonads marked by the white asterisks (*) and the dashed lines that indicate where the nuclei start transitioning into meiotic pre-phase I. (B) Quantification of the percent of mitotic nuclei in the S-phase (EdU positive) upon depletion of HSF-1 from the germline of young adults for 16 h. Mean and standard deviation are plotted (control: $N = 10$; HSF-1 depletion: $N = 9$). The P-value was calculated by unpaired t-test. (C, D) Representative images (C) and quantification (D) of acridine orange (AO) staining upon depletion of HSF-1 from the germline of young adults for 24 h. The white arrows indicate apoptotic cells at the pachytene stage. The red arrows indicate degenerative oocytes. The number of AO-positive oocytes in each gonad arm is quantified, and the mean and standard deviation are plotted ($N = 15$). The P-value was calculated by unpaired t-test. (E) Fecundity as measured by self-progenies when HSF-1 was depleted from the germline of hermaphrodites starting from Day 1 of adulthood. Mean and SEM are plotted (three experiments, $N > = 10$ for each experiment). (F) Fecundity as measured by mated progenies when HSF-1 was depleted from the germline of hermaphrodites and mated with N2 males on Day 1 of adulthood. Mean and standard deviation are plotted (control: $N = 12$; HSF-1 depletion: $N = 11$). (G) Embryo lethality caused by depletion of germline HSF-1. HSF-1 was depleted from the germline of hermaphrodites starting from Day 1 of adulthood, and embryo lethality in self-progenies and mated progenies (with N2 males) was measured on Day 2 and after. Mean and standard deviation are plotted (self-reproduction: $N = 15$; mating experiments: $N > = 10$). P-values were calculated by unpaired t-test. ns: $p > = 0.05$. Source data are available online for this figure.

germline proteome in the *daf-2(e1370)* animals rather than stress response underlies its resilience against low protein folding capacity.

Finally, we tested if reduced IIS enhances proteostasis in germ cells without HSF-1. Consistent with the expression data on the UPS components, ubiquitylated proteins did not significantly increase in the germline of *daf-2(e1370)* animals upon HSF-1 depletion, implicating less misfolded proteins resulted from HSF-1 depletion when IIS was reduced (Figs. 3F and EV3A). Although the *daf-2(e1370)* mutation was not sufficient to maintain histone protein levels in the oocytes without HSF-1 (Figs. 3G and EV3A–C), further reducing IIS by *daf-2* RNAi restored the levels of GFP::H2B (Figs. 3G and EV3C). This is consistent with the observation that HSF-1 depletion had minimal impacts on fecundity and embryo

lethality in the *daf-2(e1370)* animals treated with *daf-2* RNAi (Fig. 3A,B). Furthermore, reduced IIS, as in the *daf-2(e1370)* animals, prevented the accumulation of high molecular weight, ubiquitylated proteins in the detergent-insoluble aggregates upon HSF-1 depletion from the germline (Fig. 3H). Collectively, our results indicate that reduced IIS enhances germline proteostasis when protein folding capacity is compromised upon loss of HSF-1.

## Insulin/IGF-1 signaling (IIS) activates ribosome biogenesis and translation in germ cells

Next, we attempted to understand how reduced IIS remodels the germline proteostasis network via transcriptomic analysis in isolated germline nuclei from the *daf-2(e1370)* and the wild-type

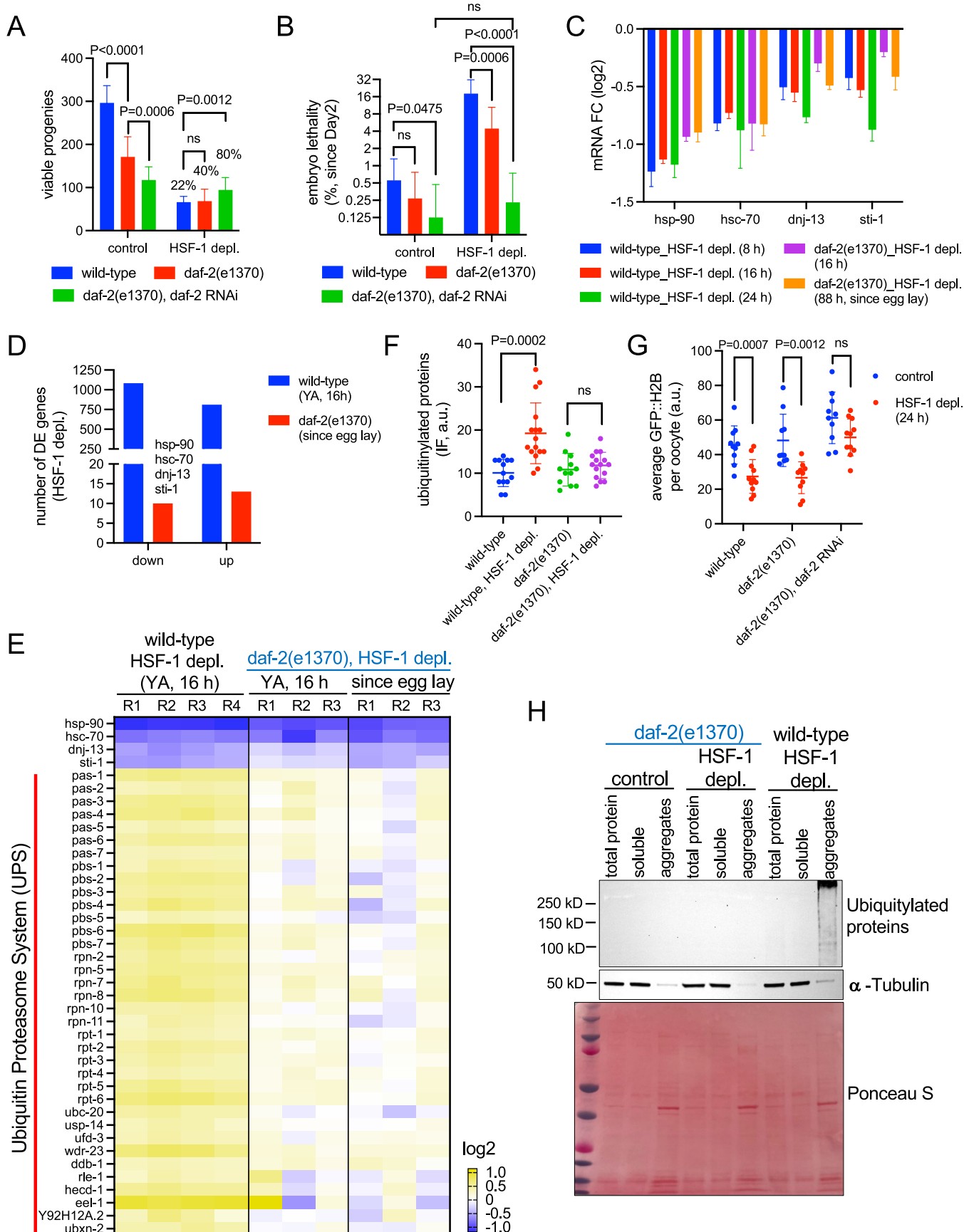

**Figure 3. Reduced Insulin/IGF-1 signaling (IIS) confers resilience against limited protein folding capacity in gametogenesis.**

(A, B) Brood size (A) and embryo lethality (B) of animals with reduced IIS upon depletion of HSF-1 from the germline starting from Day 1 of adulthood. The wild-type and *daf-2(e1370)* animals were treated with *daf-2* RNAi or control RNAi (L4440) starting at egg lay. In brood size analysis, the y-axis depicts the number of viable progenies. The percentage of brood size obtained upon HSF-1 depletion compared with the control in the same genetic background is labeled. Embryo lethality was measured on Day 2 and after. Mean and standard deviation are plotted ($N \geq 15$). *P*-values were calculated by unpaired t-test. ns: $p \geq 0.05$. (C) Histograms showing mRNA fold changes (FC) of selective chaperone and co-chaperone genes upon HSF-1 depletion from the germline as measured by RNA-seq analysis in the wild-type and *daf-2(e1370)* animals. The four chaperone and co-chaperone genes showed significant changes in mRNA levels (FDR: 0.05) upon HSF-1 depletion in the wild-type and *daf-2(e1370)* animals. In the wild-type animals, HSF-1 was depleted from the young adults for 8 h, 16 h, or 24 h. In the *daf-2(e1370)* animals, HSF-1 depletion was done in the young adults for 16 h or started from egg lay for 88 h till the animals grew to gravid adults. Mean and standard deviation are plotted (wild-type 8 h & 16 h: $N = 4$; wild-type 24 h & *daf-2(e1370)* experiments: $N = 3$). (D) Histograms showing the number of differentially expressed (DE) genes (FDR: 0.05) caused by HSF-1 depletion from the germline in the wild-type animals at young-adult (YA) stage for 16 h and from the *daf-2(e1370)* animals starting from egg lay. (E) Heatmap of mRNA fold changes at selective chaperone and co-chaperone genes as well as components of the ubiquitin-proteasome system (UPS) upon depletion of HSF-1 from the germline of the wild-type and *daf-2(e1370)* animals. Heatmap columns represent biological replicates. (F) Quantification of ubiquitylated proteins in the gonads by immunofluorescence (IF) upon germline-specific depletion of HSF-1 for 24 h from young adults of the wild-type and *daf-2(e1370)* animals. Mean and standard deviation are plotted ($N \geq 12$). *P*-values were calculated by unpaired t-test. ns: $p \geq 0.05$. (G) Quantification of protein levels of GFP::H2B transgene in fully grown oocytes upon germline-specific depletion of HSF-1 for 24 h from young adults. The wild-type and *daf-2(e1370)* animals were treated with *daf-2* RNAi or control RNAi (L4440) starting from egg lay. Mean and standard deviation are plotted ($N \geq 10$). *P*-values were calculated by unpaired t-test. ns: $p \geq 0.05$. (H) Western blot of protein aggregates upon germline-specific depletion of HSF-1 in young adults of the *daf-2(e1370)* and wild-type animals for 24 h. Antibodies against ubiquitin and α-tubulin were used. The 'aggregates' fraction was loaded as 10-fold of the other fractions. Source data are available online for this figure.

animals (Fig. EV3D). Genes involved in protein synthesis are among the most enriched functional groups that decreased expression (Fig. 4A), which include those encoding protein components of both ribosomal subunits, translation initiation and elongation factors, and one mitochondrial ribosomal protein gene (Fig. 4B). This result indicates that the translation machinery is coordinately downregulated when IIS is low. Consistent with transcriptomic data, the protein levels of endogenously tagged RPS-6::mCherry significantly decreased in the *daf-2(e1370)* germline compared to that in the wild-type control, suggesting that IIS activates ribosome biogenesis in germ cells (Fig. 4C,D). Finally, we found the global translation rate in the germline was dependent on IIS activity. Translation, as measured by O-propargyl-puromycin (OPP) incorporation, was significantly decreased in the germline of *daf-2(e1370)* animals, and further reduced upon *daf-2* RNAi (Fig. 4E,F). On the contrary, *daf-16* RNAi in the *daf-2(e1370)* animals partially restored translation (Fig. 4E,F). As DAF-16 is an essential regulator of GSC proliferation downstream of IIS (Michaelson et al, 2010; Pinkston-Gosse and Kenyon, 2007; Qi et al, 2012; Qin and Hubbard, 2015), our results strongly suggest that high IIS activity in wild-type animals may promote gametogenesis via the regulation of protein synthesis.

## Tuning down protein translation enhances germline proteostasis against the challenges of protein misfolding

On the other hand, does a lower rate of protein synthesis associated with reduced IIS render germline tolerant to limited protein folding capacity as upon HSF-1 depletion? To answer this question, we took *rsks-1(ok1255)*, the loss-of-function mutant of *C. elegans* ortholog of human p70 ribosomal S6 Kinase, as a model, which is known to reduce the translation rate (Pan et al, 2007). The *rsks-1(ok1255)* animals limited the accumulation of protein misfolding, as shown by the levels of ubiquitylated proteins in the germline (Figs. 5A and EV4A), and partially stabilized the histone H3 in the oocytes upon HSF-1 depletion (Fig. EV4A,B). Consistent with the measurements of proteostasis, the *rsks-1(ok1255)* animals displayed almost no additional defects in fecundity and gamete quality when HSF-1 was absent from the germline (Fig. 5B,C). Germline-specific RNAi against *rsks-1* phenocopied the results in the *rsks-1(ok1255)*

animals (Fig. EV4C,D), indicating the improvement of reproduction upon HSF-1 depletion was due to impairment of RSKS-1 in germ cells rather than cell-non-autonomous effects. Reducing IIS in animals with germline-specific knock-down of *rsks-1* did not further improve fecundity and had small additive effects on embryo survival in the absence of germline HSF-1 (Fig. EV4C,D), implicating IIS and RSKS-1 largely function in the same pathway to render germ cells resilient to loss of HSF-1. Our results support that a lower translation rate underlies the enhanced germline proteostasis by reduced IIS.

Since our data suggest that the primary role of HSF-1 in gametogenesis is to provide sufficient protein folding capacity, we further tested whether reduced IIS and low translation rate could grant germ cells resilience against proteotoxic stresses that challenge folding. We first exposed the animals to acute heat stress, which is known to induce protein misfolding. A brief heat shock at 34 °C for 15 min was sufficient to lead to ~30% embryo lethality from eggs laid within 4 h of the stress in the wild-type animals. The *daf-2(e1370)* and *rsks-1(ok1255)* animals, however, reduced embryo lethality after heat shock by 75% and 64%, respectively (Fig. 5D). We then treated the animals with bortezomib, which inhibits proteasome and hinders the removal of misfolded proteins, and paromomycin, which induces translation misreading, during the entire reproductive period. These two stressors significantly increased embryo lethality in the wild-type animals but failed to do so in either *daf-2(e1370)* or *rsks-1(ok1255)* animals (Fig. 5E).

Collectively, our data suggest that tuning down translation by reduced IIS provides germline resilience against the challenges of protein misfolding either due to insufficient chaperone expression or acute and chronic proteotoxic stresses. Conversely, IIS-stimulated protein synthesis requires HSF-1-dependent protein folding capacity to maintain germline proteostasis.

## PEPT-1 functions downstream from insulin/IGF-1 signaling (IIS) and FOXO/DAF-16 to determine the requirement for HSF-1 in germline development

Our previous study showed that the requirement for HSF-1 in germline development is dictated by DAF-16 activity in the soma

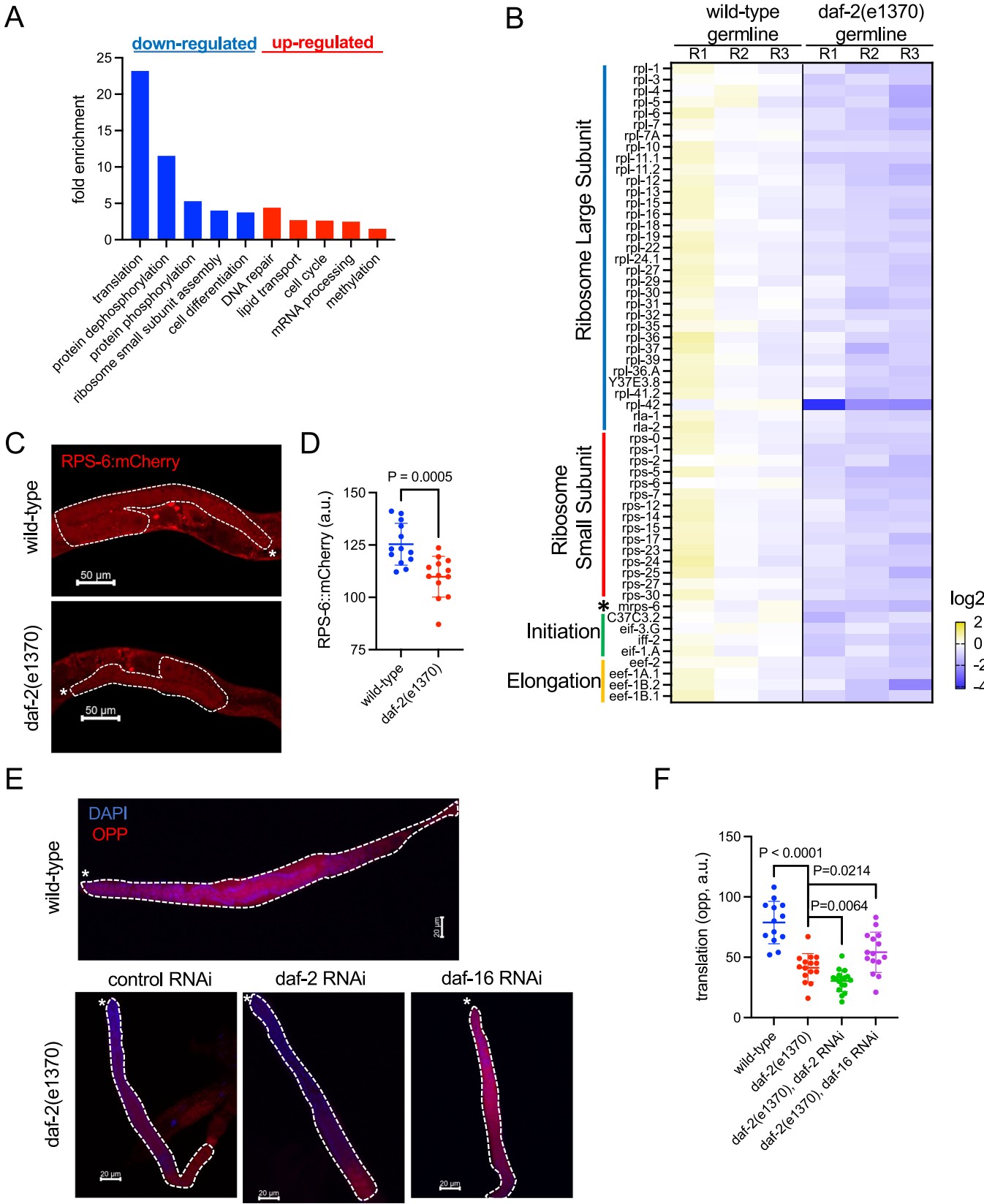

◄ **Figure 4.   Insulin/IGF-1 signaling (IIS) activates ribosome biogenesis and translation in germ cells.**

(A) Gene Ontology (GO) analyses of differentially expressed (DE) genes (FDR: 0.05) when comparing the germline of the *daf-2(e1370)* animals to that in the wild-type animals. RNA-seq analysis was performed in isolated germline nuclei from young adults of the wild-type and *daf-2(e1370)* animals (*N* = 3). The top 5 GO terms based on enrichment scores are shown for downregulated genes (blue bars) and upregulated genes (red bars) in the *daf-2(e1370)* germline. (B) Heatmap of the mRNA levels of translation-related genes in the wild-type and *daf-2(e1370)* germline. The fold change against the average of mRNA in the wild-type is shown. Genes that function in translation based on GO analysis and show significant changes in the *daf-2(e1370)* germline (FDR: 0.05) are included. The mitochondrial ribosomal protein gene, mrps-6, is labeled by *. Heatmap columns represent biological replicates. (C, D) Representative images (C) and quantification (D) of protein levels of the endogenously tagged RPS-6::mCherry in the gonad of wild-type and *daf-2(e1370)* animals. The dashed lines outline the gonads, and the white asterisks (*) indicate the distal end of the gonads. Mean and standard deviation are plotted (*N* = 13). The *p*-value was calculated by unpaired t-test. (E&F) Representative images (E) and quantification (F) of protein translation in the germline of animals with altered IIS. Newly synthesized proteins were measured by O-propargyl-puromycin (OPP) incorporation. The wild-type and *daf-2(e1370)* animals were treated with *daf-2* RNAi, *daf-16* RNAi, or control RNAi starting from egg lay. Mean and standard deviation are plotted for quantification (*N* > = 13). *P*-values were calculated by unpaired t-test. Source data are available online for this figure.

(Edwards et al, 2021). Since DAF-16 is involved in the regulation of germline protein synthesis by IIS (Fig. 4E,F), we hypothesize that DAF-16, through its transcriptional target genes in somatic tissues non-cell-autonomously impacts protein synthesis and proteostasis in the germline. To identify in which tissue DAF-16 mediates this non-cell-autonomous regulation, we performed tissue-specific RNAi against DAF-16 in the *daf-2(e1370)* animals and tested the sensitivity of germline development to HSF-1 depletion starting at egg lay (Fig. 6A). As we previously reported, the *daf-2(e1370)* animals could reproduce without germline HSF-1 through larval development, while systemic RNAi against *daf-16* led to sterility. The *daf-16* RNAi in the muscle or intestine significantly decreased the brood size of the *daf-2(e1370)* animals upon HSF-1 depletion but did not cause complete sterility, suggesting that DAF-16 functions from both tissues to regulate germline proteostasis. In contrast, *daf-16* RNAi in the hypodermis or germline did not significantly alter the brood size of the *daf-2(e1370)* animals upon HSF-1 depletion.

Based on transcriptomic data and DNA motif analysis, the DAF-16-responsive genes in the IIS pathway have been grouped into two classes: DAF-16 directly activates Class I genes through the DAF-16-binding element (DBE) and indirectly represses Class II genes that contain the DAF-16-associated element (DAE) (Tepper et al, 2013). We decided to focus on a top-ranked Class II gene, *pept-1*, in our study for the following reasons: first, it encodes an oligopeptide transporter specifically expressed in the intestine, one primary tissue from which DAF-16 regulates germline proteostasis; second, the genetic interactions of PEPT-1 with IIS and mTORC1, two significant regulators of protein synthesis, have been reported (Benner et al, 2011; Geillinger et al, 2014; Meissner et al, 2004); finally, we showed a loss-of-function mutant, *pept-1*(lg601) was sufficient to rescue reproduction in the absence of germline HSF-1 during entire larval development and adulthoods as reduced IIS does (Fig. 6B).

Our qPCR analysis showed that the expression of *pept-1* decreased in the *daf-2(e1370)* animals and was restored in the *daf-2(e1370); daf-16(mgDf50)* double mutant (Fig. 6C), confirming that IIS activates the expression of *pept-1* by alleviating the repression by DAF-16. In contrast, *sod-3*, a Class I gene, was activated by DAF-16 when IIS was reduced (Fig. 6D). Supporting the notion that PEPT-1 functions downstream of DAF-16 in regulating germline proteostasis, inactivation of PEPT-1 partially restored fertility in the *daf-2(e1370); daf-16(mgDf50)* double mutant when HSF-1 was depleted from the germline starting at egg lay (Fig. 6E). These results strongly suggest that PEPT-1 plays a vital role in the germline proteostasis pathway mediated by IIS and DAF-16.

## Insulin/IGF-1 signaling (IIS) regulates germline protein synthesis and proteostasis via peptide uptake in the intestine

PEPT-1 is located at the apical membrane of the intestine and is responsible for the uptake of di-/tri-peptides from dietary proteins (Spanier, 2014). We then tested if IIS regulates germline protein synthesis via protein absorption in the intestine. Consistent with published data, RNAi against *pept-1* significantly impaired peptide uptake into the intestine (Fig. 7A,B). Similar results were observed in animals with reduced IIS. In particular, the *daf-2(e1370)* animals treated with *daf-2* RNAi significantly decreased peptide uptake (Fig. 7C,D). Furthermore, *pept-1* RNAi was sufficient to reduce the RPS-6 levels and translation rate in the germline, suggesting PEPT-1-mediated peptide uptake non-cell-autonomously promotes ribosome biogenesis and protein synthesis during gametogenesis (Figs. 7E,F and EV5A,B). Supporting the importance of dietary regulation on germline proteostasis, the *eat-6(ad467)* animals that have decreased food uptake due to defects in pharyngeal feeding (Avery, 1993) also reduced germline protein synthesis (Fig. EV5C,D) and partially rescued reproduction (12 out of 15 animals) without HSF-1 in germ cells during larval development and adulthood (Fig. 7G). Finally, we tested if the change in peptide uptake impacts germline proteostasis by altering the pool of amino acids available for translation. A published study has shown that almost all free amino acids were decreased in the cytosol in the *pept-1(lg601)* animals compared to that in the wild-type (Geillinger et al, 2014). Supplement of a mixture of amino acids on the culture plates of *pept-1 (lg601)* animals has increased the number of mated progenies by 20% (P:0.06) in the presence of HSF-1 but significantly decreased the brood size upon HSF-1 depletion (Fig. 7H). Similar results were obtained when we measured self-progenies (Fig. EV5E). These results suggest that loss of PEPT-1 limits protein absorption and, consequently, the amino acid pool used for germline protein synthesis, which lowers the translation rate, rendering germline development less dependent on HSF-1-mediated protein folding.

## Discussion

Proteostasis is the cellular state in which protein synthesis, folding, transport, and turnover are well coordinated to maintain a functional and dynamic proteome (Balch et al, 2008). Gametogenesis involves periods of very active protein synthesis and is particularly dependent on proteostasis (Cafe et al, 2021; Muhammad and Li, 2023). On the other hand, gametogenesis is highly energy-demanding and sensitive

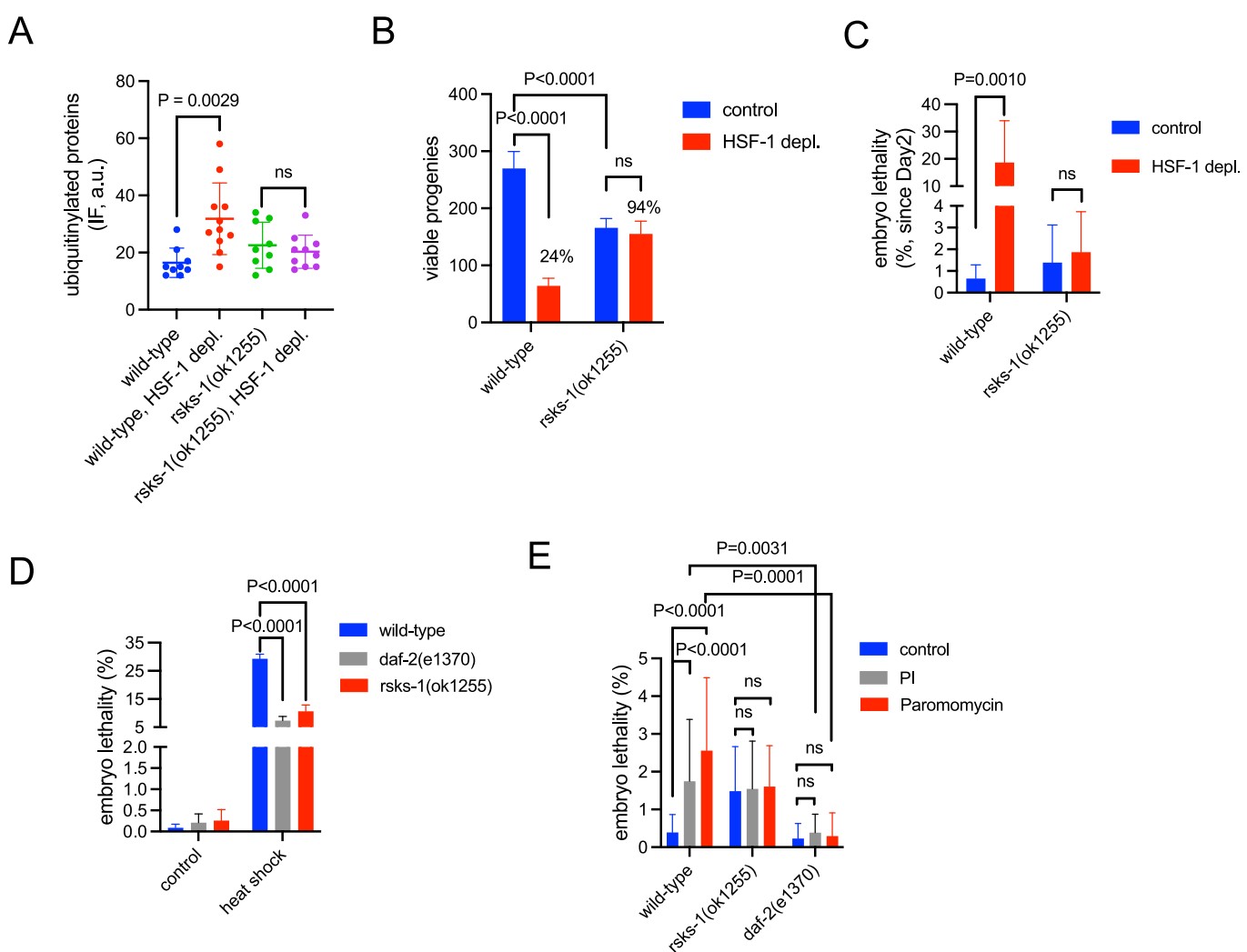

**Figure 5. Reduced translation rate underlies robust germline proteostasis and resilience against proteotoxic stress mediated by low insulin/IGF-1 signaling (IIS).**

(A) Quantification of ubiquitylated proteins in the gonads by immunofluorescence (IF) upon germline-specific depletion of HSF-1 for 24 h from young adults of the wild-type and *rsks-1(ok1255)* animals. Mean and standard deviation are plotted ($N \geq 9$). *P*-values were calculated by unpaired t-test. ns: $p > 0.05$. (B, C) Brood size (B) and embryo lethality (C) of the *rsks-1(ok1255)* animals upon depletion of HSF-1 from the germline starting from Day 1 of adulthood. In brood size analysis, the y-axis depicts the number of viable progenies. The percentage of brood size obtained upon HSF-1 depletion compared with the control in the same genetic background is labeled. Embryo lethality was measured on Day 2 and after. Mean and standard deviation are plotted (wild-type control, $N = 10$; wild-type HSF-1 depl., $N = 20$; *rsks-1(ok1255)* control, $N = 24$; *rsks-1(ok1255)* HSF-1 depl., $N = 18$). *P*-values were calculated by unpaired t-test. ns: $p \geq 0.05$. (D) Embryo lethality caused by acute heat stress in the wild-type, *daf-2(e1370)* and *rsks-1(ok1255)* animals. Young gravid adults were heat-shocked briefly at 34 °C for 15 min, and embryo lethality was measured in eggs laid within 4 h of the recovery at 20 °C. Experiments were done in 8 biological replicates, each containing 5 animals. Mean and SEM are plotted. *P*-values were calculated by unpaired t-test. ns: $p > 0.05$. (E) Embryo lethality caused by the proteasome inhibitor bortezomib (PI) and paromomycin that induces translational misreading in the wild-type, *daf-2(e1370)* and *rsks-1(ok1255)* animals. Animals were treated with either 10 μM of bortezomib or 1 mM of paromomycin starting at the young adult stage. Embryo lethality through the reproductive period was measured. Mean and standard deviation are plotted ($N \geq 15$). *P*-values were calculated by unpaired t-test. ns: $p \geq 0.05$. Source data are available online for this figure.

to energy metabolism and environmental stress, which impact both genome and proteome stability. Compared to the extensively studied mechanisms that safeguard germline genome integrity, how animals regulate germline proteostasis at the organismal level is still poorly understood. In this study, we took *C. elegans* as a model and demonstrated that HSF-1-dependent protein folding capacity needs to be coupled with the rate of protein synthesis controlled by insulin/IGF-1 signaling (IIS) to achieve germline proteostasis. Especially, IIS activates the expression of the intestinal peptide transporter, *pept-1*, by alleviating its repression from DAF-16, which provides a non-cell-

autonomous mechanism that controls germline ribosome biogenesis and translation via regulation of dietary protein absorption (Fig. 7I).

## HSF-1 works in concert with the insulin/IGF-1 signaling (IIS) pathway to provide sufficient protein folding capacity in germ cells

HSF-1 is best known for its ability to respond to proteotoxic stress such as heat shock and induce the expression of a series of protein quality control factors, including molecular chaperones,

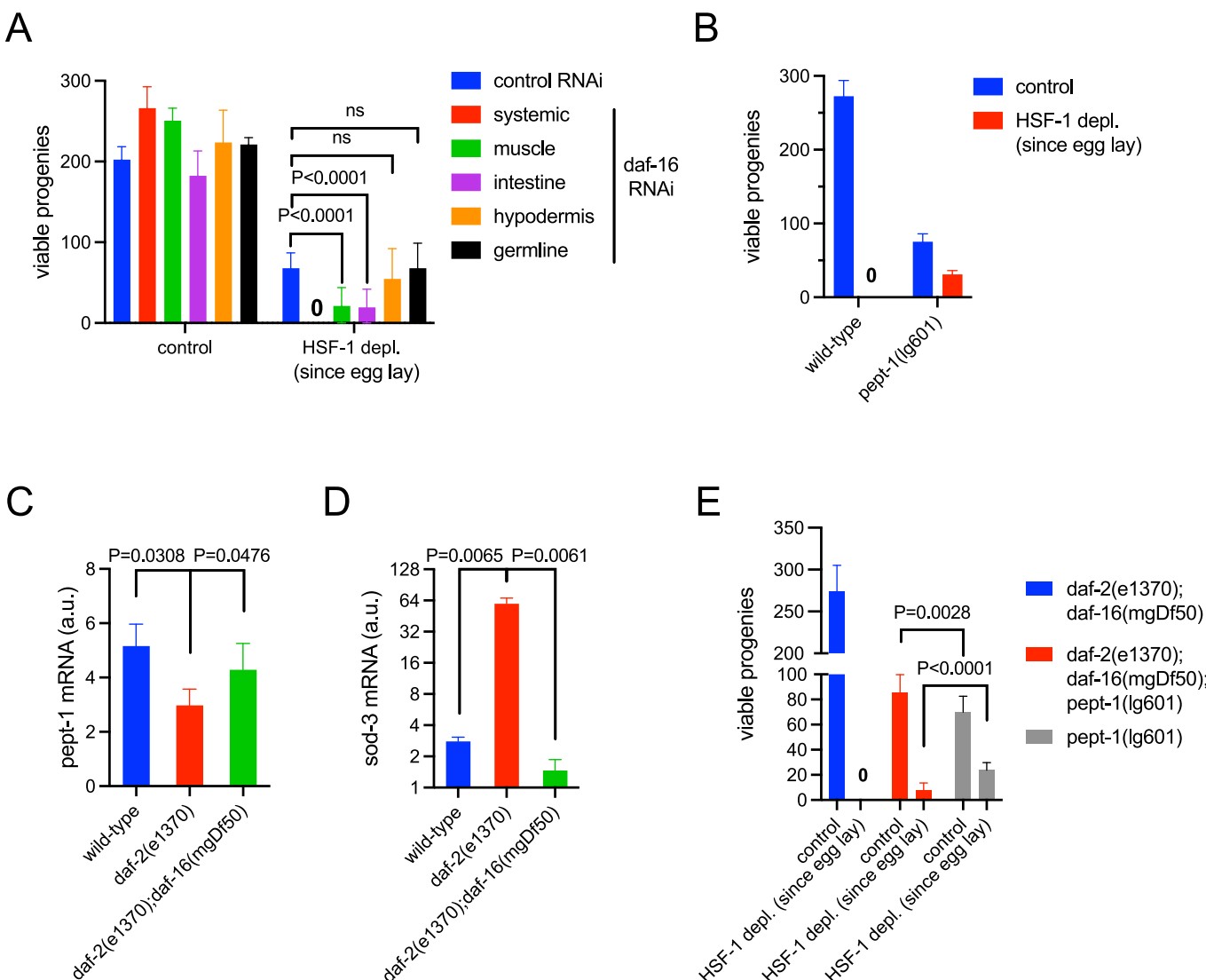

**Figure 6. Intestinal peptide transporter PEPT-1 functions downstream from FOXO/DAF-16 to dictate the requirement for HSF-1 in germline development.**

(A) Brood size of the *daf-2(e1370)* animals treated with *daf-16* RNAi upon depletion of HSF-1 from the germline starting from egg lay. The *daf-2(e1370)* animals were treated with control RNAi (L4440) or *daf-16* RNAi either in the whole animal (systemic) or in specific tissues starting from egg lay. Mean and standard deviation are plotted (N > = 14). P-values were calculated by unpaired t-test. ns: p > =0.05. (B) Brood size of the loss-of-function mutant *pept-1(lg601)* upon depletion of HSF-1 from the germline starting from egg lay. Mean and standard deviation are plotted (N > = 14). (C, D) The RT-qPCR analysis of *pept-1* (C) and *sod-3* (D) in the wild-type, *daf-2(e1370)* and *daf-2(e1370);daf-16(mgDf50)* animals at late L4 stage. Mean and standard deviation of biological triplicates are plotted. P-values were calculated by paired t-test. (E) Brood size of the *pept-1(lg601)*, *daf-2(e1370);daf-16(mgDf50)* and *daf-2(e1370);daf-16(mgDf50); pept-1(lg601)* animals upon depletion of HSF-1 from the germline starting from egg lay. Mean and standard deviation are plotted (N > = 14). P-values were calculated by unpaired t-test. Source data are available online for this figure.

detoxification enzymes, and players in protein clearance pathways to restore proteostasis (Li et al, 2017). Published works from us and colleagues have shown that only a subset of germ cells can induce the classical heat shock response (HSR) upon temperature stress (Das et al, 2020; Edwards et al, 2021). Instead, HSF-1 promotes the expression of selective chaperones and co-chaperones that are important for the folding and maturation of nascent proteins in physiological conditions (Edwards et al, 2021). Although HSF-1 also binds and activates other genes in germ cells, our data suggest that HSF-1's primary role in gametogenesis is to provide protein folding capacity. Loss of HSF-1 led to protein misfolding and

degradation, which over time exhausted the UPS, causing the accumulation of protein aggregates (Fig. 1).

Interestingly, the requirement for HSF-1 in gametogenesis is determined by the activity of IIS. Using a reduction-of-function mutant of the IGF-1R/DAF-2 and RNAi, we found the defects in fecundity, gamete quality as well as proteostasis caused by HSF-1 depletion negatively correlated with IIS activity (Fig. 3). As IIS increases the translation rate in the germline (Fig. 4E), we propose that HSF-1-dependent chaperone expression must be coupled with IIS-activated protein synthesis to ensure sufficient capacity for nascent folding and protein maturation during gametogenesis. The

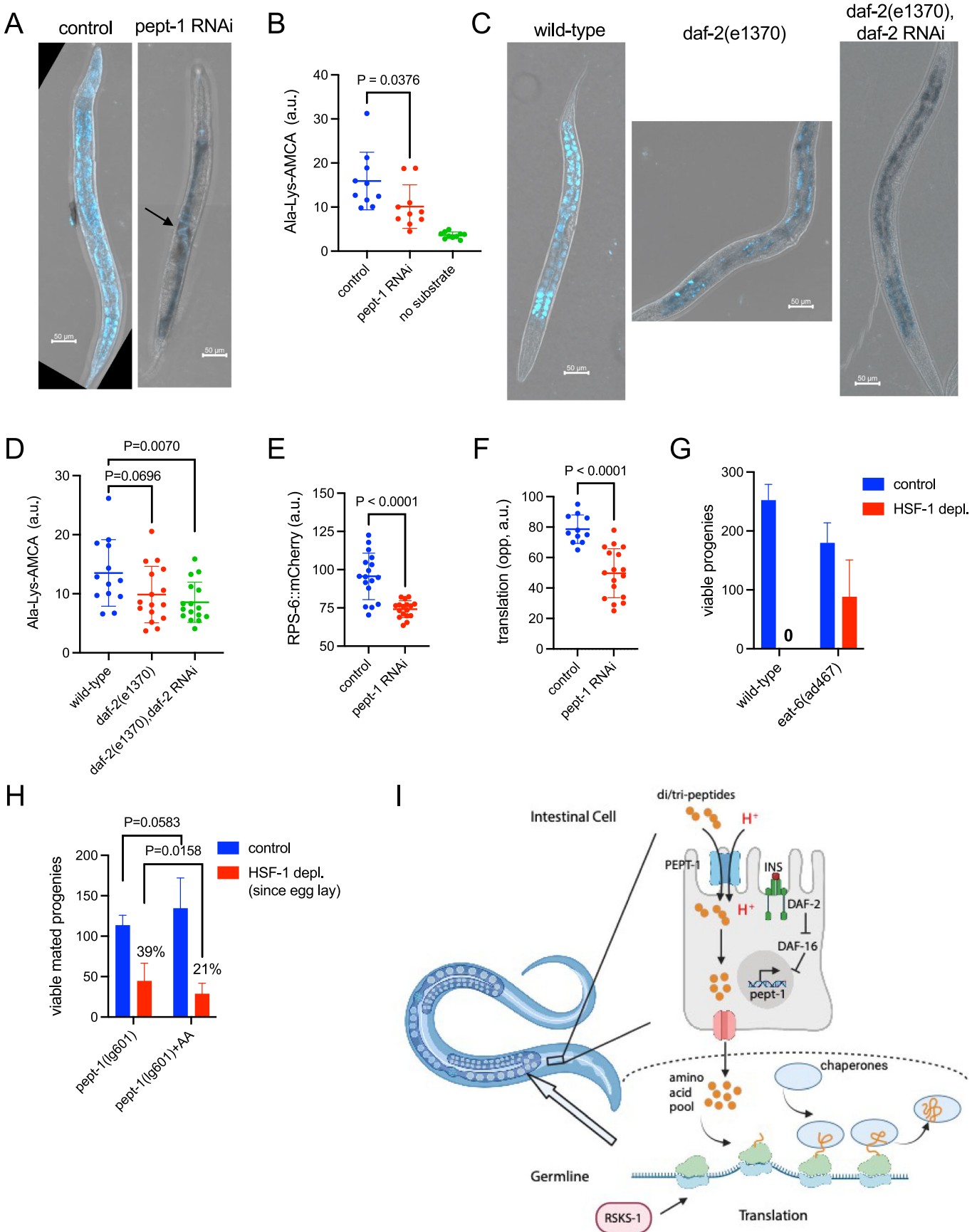

**Figure 7.   Insulin/IGF-1 signaling (IIS) regulates germline protein synthesis and proteostasis via peptide uptake in the intestine.**

(A, B) Representative images (A) and quantification (B) of intestinal peptide uptake measured by Ala-Lys-AMCA upon *pept-1* RNAi. RNAi started at egg lay, and peptide uptake was measured in late L4 and young adult stages. The signal in the *pept-1* RNAi image largely came from Ala-Lys-AMCA trapped in the lumen (as pointed by the arrow). Mean and standard deviation are plotted ($N = 10$). The *P*-value was calculated by unpaired t-test. (C, D) Representative images (C) and quantification (D) of intestinal peptide uptake in animals with reduced IIS. Experiments were done as in (A, B). The wild-type animals treated with control RNAi were used as the positive control. Mean and standard deviation are plotted (wild-type: $N = 13$; *daf-2(e1370)* experiments: $N = 16$). *P*-values were calculated by unpaired t-test. (E, F) Quantification of the endogenously tagged RPS-6::mCherry protein (E) and translation measured by OPP incorporation (F) in the germline upon *pept-1* RNAi. RNAi started at egg lay, and measurement was done in young adults. Mean and standard deviation are plotted (RPS-6 experiments: $N = 17$; for translation assays, control: $N = 10$, *pept-1* RNAi: $N = 17$). *P*-values were calculated by unpaired t-test. (G) Brood size analysis of the *eat-6(ad467)* animals upon depletion of HSF-1 from the germline starting from egg lay. Mean and standard deviation are plotted ($N > 14$). (H) Brood size analysis of the *pept-1(lg601)* animals in the presence or absence of amino acid (AA) supplements. Both depletion of HSF-1 from the germline and the supplement of AA started from egg lay. The *pept-1(lg601)* animals were mated with N2 males on Day 1 of adulthood. Mean and standard deviation are plotted ($N > 12$). *P*-values were calculated by unpaired t-test. The percentage of brood size obtained upon HSF-1 depletion compared with the corresponding control is labeled. (I) Model for non-cell-autonomous regulation of germline proteostasis by insulin/IGF-1 signaling (IIS). IGF-1R/DAF-2 activates the expression of the *pept-1* gene by alleviating its repression by FOXO/DAF-16, which promotes the uptake of di/tri-peptides from the dietary proteins. This pathway non-cell-autonomously increases the pool of amino acids that fuel ribosomal biogenesis and translation in the germline, which is the primary site of protein synthesis in reproductive animals. HSF-1-dependent chaperone expression must be at a delicate balance with IIS activity, providing sufficient folding capacity for nascent proteins. Reduced IIS lowers ribosomal biogenesis and translation, rendering the germline resilient to limited protein folding capacity and proteotoxic stress. Source data are available online for this figure.

coupling mechanism is likely through the regulation of HSF-1 transcriptional activity. Our previous work has shown that essential chaperone genes like *hsc-70* and *hsp-90* can be expressed at the basal level without HSF-1, but HSF-1 tunes up their expression in response to IIS (Edwards et al, 2021). This explains why only a small subset of HSF-1-dependent chaperone genes (*hsp-90*, *hsc-70*, *dnj-13*, and *sti-1*) decreased expression upon HSF-1 depletion in the *daf-2(e1370)* mutant (Fig. 3D). These four genes showed the most prominent fold-change upon HSF-1 depletion in the wild-type animals (Fig. 1A, 16 h time point). The changes on the other chaperone genes likely fell below detection sensitivity with the lower HSF-1 activity in the germline of animals with reduced IIS. However, our previous work did not answer how IIS stimulates HSF-1 activity in germ cells. In this study, we found that high IIS activity, as in wild-type animals, promotes germline protein synthesis, providing a plausible mechanism. It is known that chaperones, including HSC-70, HSP-90, and the TRiC/CCT complex, interact with HSF-1 and inhibit its activity (Neef et al, 2014; Shi et al, 1998; Zou et al, 1998). When IIS activity is high, the consequent high influx of nascent proteins titrates chaperones from HSF-1, releasing HSF-1 from the inhibitory state to upregulate chaperone expression and augment protein folding capacity. Coupling IIS and HSF-1 activities provides a strategy to ensure sufficient chaperone expression for nascent folding. This mechanism may have additional meanings in gametogenesis as molecular chaperones have evolved specialized roles in meiosis. For example, in *C. elegans*, IIS and HSP-90 are required for MAPK activation during oocyte growth (Green et al, 2011; Lopez et al, 2013). In future studies, it is interesting to determine how HSF-1-dependent *hsp-90* expression may contribute to IIS-MAPK signaling in gametogenesis.

### The low rate of ribosome biogenesis and translation underlies robust germline proteostasis mediated by reduced insulin/IGF-1 signaling (IIS)

Animals with reduced IIS were more resilient to proteotoxic challenges associated with loss of HSF-1 from the germline (Fig. 3F,G) and by acute and chronic stresses (Fig. 5D,E). It is well-established that reduced IIS improves proteostasis and extends

lifespan by enhancing stress responses in somatic tissues (Murphy et al, 2003). This is not likely the primary mechanism underlying the robust germline proteostasis by reduced IIS. First, although the expression of HSC-70 and HSP-90 were significantly down-regulated in the *daf-2(e1370)* animals upon chronic HSF-1 depletion (Fig. 3C), it did not induce a stress-response signature based on our transcriptomic analysis (Fig. 3D,E). Second, the two stress-responsive factors downstream of IIS, NRF/SKN-1 and FOXO/DAF-16 were not required from the germline by reduced IIS for reproduction in the absence of HSF-1 (Edwards et al, 2021). NRF/SKN-1 activates the antioxidant response (Tullet et al, 2008) and stress response against proteasome dysfunction (Lehrbach et al, 2019), and FOXO/DAF-16 promotes the expression of chaperones, UPS components, and other protein quality control genes upon stress (Henis-Korenblit et al, 2010; Hsu et al, 2003; Sun et al, 2020; Vilchez et al, 2012). Most importantly, lowering the translation rate in the *rsks-1(ok1255)* mutant phenocopied the robust germline proteostasis observed in animals with reduced IIS when challenged by HSF-1 depletion or stresses (Fig. 5). These data provide compelling evidence that tuning down protein synthesis by reducing IIS confers resilience against limited protein folding capacity and proteotoxic stresses in gametogenesis.

Our finding is consistent with the role of IIS in nutrient and stress sensing and with the previous findings of IIS in larval development (Murphy and Hu, 2013). In favorable conditions, IIS is high, speeding up development and reproduction, while under adverse conditions, IIS decreases, promoting survival and stress resilience. However, our findings suggest that the somatic tissues and the germline utilize different strategies to enhance proteostasis upon IIS reduction. The germline is the primary site of protein synthesis in reproductive animals. Therefore, tuning down ribosome biogenesis and translation could be sufficient to slow down the protein life cycle and spare chaperones and protein clearance pathways to cope with damaged and misfolded proteins upon stress. In addition, meiotic cells in the diakinesis state undergo chromosome compaction and transcription repression, limiting transcriptional stress responses. Conversely, somatic tissues may require the induction of additional protein quality control factors through transcriptional responses to cope with proteotoxic stresses. Our RNA-seq data suggest that reduced IIS down-regulates almost

all the ribosomal protein genes and multiple factors that function at translation initiation and elongation (Fig. 4B). Future studies will determine if tuning down protein synthesis coordinately at ribosome biogenesis and both initiation and elongation steps of translation is important for enhanced proteostasis.

Interestingly, reduced IIS is known to improve oocyte quality in maternal aging, which requires the activity of DAF-16 non-cell-autonomously from the muscle and the intestine, the same tissues where DAF-16 is required for dictating the resilience against HSF-1 depletion from the germline (Fig. 6A). Loss of proteostasis is a hallmark of aging, so reduced IIS may delay oocyte aging by enhancing germline proteostasis. Future studies will determine if PEPT-1 has a role in reproductive aging and identify the signal of DAF-16 from the muscle that regulates germline proteostasis. Notably, IIS can also cell-autonomously impact germline proteostasis. A previous study shows that IIS regulates oocyte quality in aging via the Cathepsin B protease activity (Templeman et al, 2018).

### Transcriptional control of *pept-1* by Insulin/IGF-1 signaling (IIS) contributes to its non-cell-autonomous regulation of germline proteostasis

Based on the following evidence, our work suggests that the intestinal oligo-peptide transporter, PEPT-1, is an important mediator for IIS to regulate protein synthesis and proteostasis in the germline. First, IIS activates the expression of *pept-1* by alleviating its repression from DAF-16 (Fig. 6C). The *pept-1* promoter contains DAE (DAF-16-associated element), the proposed binding site of PQM-1, a transcription factor whose activity is antagonized by DAF-16 (Tepper et al, 2013). The ModEncode ChIP-seq data suggest that PQM-1 binds to the *pept-1* promoter (Niu et al, 2011), but its role in regulating *pept-1* expression is yet to be determined. Second, the loss-of-function mutant *pept-1(lg601)* partially rescued reproduction in both the wild-type and *daf-2(e1370); daf-16(mgDf50)* double mutant when HSF-1 was absent from the germline through larval development (Fig. 6B,E) and was sufficient to reduce ribosome biogenesis and translation rate in the germline (Fig. 7E,F). It argues that PEPT-1 functions downstream from IIS and DAF-16 to regulate germline proteostasis. Finally, reduced IIS decreased the peptide uptake (Fig. 7C,D). Dietary proteins are hydrolyzed to oligopeptides in the intestinal lumen and further processed into di-/tri-peptides and free amino acids to enter enterocytes via separate transporters (Spanier, 2014). *C. elegans* PEPT-1 is the ortholog of the low-affinity, high-capacity di-/tri-peptide transporter PEPT1 in mammals (Fei et al, 1998), which serves as one major protein absorption pathway. Supplementation of a mixture of amino acids in the culture increased the brood size of the *pept-1(lg601)* animals by ~20% in the presence of HSF-1 (Figs. 7H and EV5E), which is consistent with previous results (Meissner et al, 2004; Spanier et al, 2018) as increasing the uptake of free amino acids through the corresponding transporters would not completely compensate for the loss of oligo-peptide uptake by PEPT-1 in protein absorption. However, this was sufficient to increase the sensitivity of the *pept-1(lg601)* animals to HSF-1 depletion from the germline (Figs. 7H and EV5E). Although we cannot rule out the possibility that amino acid supplementation may impact germline proteostasis independently of providing substrates for translation, it is reasonable to think that IIS through

PEPT-1 promotes dietary protein absorption, which is subsequently incorporated into a pool of amino acids to fuel germline protein synthesis (Fig. 7I).

It has been reported that PEPT-1 genetically interacts with IIS and mTORC1 pathways to regulate development, reproduction, and longevity (Benner et al, 2011; Geillinger et al, 2014; Meissner et al, 2004). However, those studies have focused on the intestinal cells or the organismal functions. As the germline is the primary site of protein synthesis in reproductive animals and amino acids stimulate mTORC1 activity via multiple signaling pathways (Shimobayashi and Hall, 2016), we propose that PEPT-1-dependent amino acid sufficiency could regulate mTORC1 in the germline and impact translation. Our results on RSKS-1, the critical regulator of translation downstream of mTORC1 (Blackwell et al, 2019; Saxton and Sabatini, 2017), support this hypothesis. We found that the *rsks-1(ok1255)* animals and germline-specific RNAi against *rsks-1* phenocopied the reproductive behaviors of animals with reduced IIS in response to HSF-1 depletion (Figs. 5 and EV4), linking mTORC1 to IIS-regulated germline proteostasis. As proteostasis is safeguarded by a network of quality control pathways, future work will test the roles of the other branches of the proteostasis network in germline proteostasis, such as autophagy, which also functions downstream of mTORC1 (Saxton and Sabatini, 2017).

PEPT-1 is a rheogenic $H^+$-dependent carrier that causes an influx of proton during transport, and it requires the sodium-proton antiporter NHX-2 to recover from the intracellular acid load (Nehrke, 2003). NHX-2 also promotes free fatty acid uptake; thus, PEPT-1's activity impacts fatty acid metabolism and storage (Spanier et al, 2009). A recent study indicates that the residency of PEPT-1 transporter at the plasma membrane is regulated by lipid homeostasis in the intestine (Watterson et al, 2022). Our work opens the opportunity for future research to understand how fatty acid and protein metabolisms interact with the glucose-sensing IIS to regulate germline proteostasis at the organismal level.

## Methods

### Reagents and tools table

| Reagent or Resource | Source | Identifier |
|---|---|---|
| **Antibodies** | | |
| Mouse monoclonal anti-ubiquitinylated proteins (FK2) | Sigma-Aldrich | Sigma-Aldrich Cat# 04-263; RRID:AB_612093 |
| Mouse monoclonal anti-α-Tubulin | Sigma-Aldrich | Sigma-Aldrich Cat# T5168, RRID:AB_477579 |
| Rabbit polyclonal anti-Histone H3 | Abcam | Abcam Cat# ab1791, RRID:AB_302613 |
| Mouse monoclonal anti-ubiquitin | Novus | Novus Cat# NB300-130, RRID:AB_2238516 |
| Goat anti-Mouse IgG (H + L) Alexa Fluor Plus 647 | Thermofisher Scientific | Invitrogen Cat# A32728; RRID:AB_2633277 |
| Goat anti-Rabbit IgG (H + L) Alexa Fluor Plus 488 | Thermofisher Scientific | Invitrogen Cat# A32731; RRID:AB_2633280 |

| Reagent or Resource | Source | Identifier |
|---|---|---|
| Goat anti-Mouse IgG (H + L) Cross-Adsorbed Secondary Antibody, HRP | Thermofisher Scientific | ThermoFisher Scientific Cat# A16072, RRID:AB_2534745 |
| **Bacterial and virus strains** | | |
| *E. coli* OP50 | CGC | OP50 |
| *E. coli* MG1693 | CGSC | MG1693 |
| **Biological samples** | | |
| **Chemicals, peptides, and recombinant proteins** | | |
| Indole-3-acetic acid (auxin) | Sigma-Aldrich | Cat# I2886-5G; Cas# 87-51-4 |
| IPTG | Thermofisher Scientific | Cat# R0393; Cas# 367-93-1 |
| Trizol reagent | Thermofisher Scientific | Cat #15596026 |
| EdU (5-ethynyl-20-deoxyuridine) | Thermofisher Scientific | Cat # A10044 |
| Paraformaldehyde | Fisher Scientific | Cat# AA433689M; Cas# 30525-89-4 |
| iTaq Universal SYBR Green Supermix | BioRad | Cat# 1725124 |
| NEBuilder HiFi DNA Assembly Master Mix | New England BioLabs | Cat# E2621 |
| Alt-R HiFi S.p. Cas9 NLS | IDT | Cat# 1078727 |
| OPP (O-propargyl-puromycin) | Thermofisher Scientific | Cat# C10459 |
| MEM Non-essential Amino Acid Solution (100×) | Sigma-Aldrich | Cat# M7145 |
| MEM Amino Acids (50×) solution without L-glutamine | Sigma-Aldrich | Cat# M5550 |
| N,N-Dimethylformamide | Sigma-Aldrich | Cat# 270547; Cas# 68-12-2 |
| β-Ala-Lys(AMCA) | Peptide Institute | Cat# 3412-v |
| cOmplete™, Mini Protease Inhibitor Cocktail | Roche | Cat# 4693124001 |
| PUREBLU™ DAPI | Biorad | Cat# 1351303 |
| **Critical commercial assays** | | |
| O-Click-iT™ Plus OPP Alexa Fluor™ 647 Protein Synthesis Assay Kit | Thermofisher Scientific | Cat# C10458 |
| Click-iT EdU Cell Proliferation Kit for Imaging, Alexa Fluor 594 dye | Thermofisher Scientific | Cat# C10339 |
| iScript cDNA synthesis kit | BioRad | Cat# 1708891 |
| Zymo Research Direct-zol RNA MiniPrep kit | Zymo Research | Cat # R2051 |
| ZymoPURE Plasmid Miniprep Kit | Zymo Research | Cat # R4208T |
| QIAGEN MinElute PCR Purification Kit | QIAGEN | Cat# 28004 |
| Pierce BCA Protein Assay Kit | Thermofisher Scientific | Cat# 23227 |

| Reagent or Resource | Source | Identifier |
|---|---|---|
| **Deposited data** | | |
| RNA-seq analysis of enriched germline nuclei from both the wild-type and daf-2(e1370) animals | This Study | GEO: GSE256186 |
| RNA-seq analysis of HSF-1 depletion from the germline of daf-2(e1370) animals | This Study | GEO: GSE256186 |
| RNA-seq analysis of HSF-1 depletion from the germline of wild-type animals | Edwards et al, 2021 | GEO: GSE162066 |
| **Experimental models: Cell lines** | | |
| **Experimental models: Organisms/strains** | | |
| Wild type *C. elegans*, Bristol | CGC | N2 |
| *mkcSi13 II; rde-1(mkc36) V* | CGC | DCL569 |
| *rde-1(ne300) V; neIs9 X* | CGC | WM118 |
| *rde-1(ne219) V; kzIs9* | CGC | NR222 |
| *frSi17 II; frIs7 IV; rde-1(ne300) V* | CGC | IG1839 |
| *daf-2(e1370)III* | CGC | CB1370 |
| *pept-1(lg601) X* | CGC | BR2742 |
| *daf-16(mgDf50) I* | CGC | GR1307 |
| *rsks-1(ok1255) III* | CGC | RB1206 |
| hsf-1(ljt3[hsf-1::degron::gfp]) I;unc-119(ed3)III;ieSi38[sun-1p::TIR1::mRuby::sun-1_3'UTR+Cbr-unc-119( + )] IV | Edwards et al, 2021 | JTL621 |
| hsf-1(ljt5[hsf-1::degron::3xFLAG])I; unc-119(ed3)III;ieSi38[sun-1p::TIR1::mRuby::sun-1_3'UTR +Cbr-unc-119( + )]IV | This study | JTL716 |
| *rps-6(rns6[rps-6::mCherry]) I* | CGC | JAR16 |
| **Oligonucleotides** | | |
| Primers used in qRT-PCR (see Method details) | This study | N/A |
| RNAi clones (multiple) | Ahringer Library; Bioscience | Cat# 3318_Cel_RNAi_complete |
| **Recombinant DNA** | | |
| hsf-1 c-ter::degron::3xFLAG in pUC19 | This study | N/A |
| **Software and algorithms** | | |
| Prism 10 | GraphPad Software | https://www.graphpad.com:443/ |
| RNA STAR 2.6 | Dobin et al, 2013 | https://github.com/alexdobin/STAR |
| Rsubread | Liao et al, 2019 | https://bioconductor.org/packages/release/bioc/html/Rsubread.html |

| Reagent or Resource | Source | Identifier |
|---|---|---|
| edgR | Robinson et al, 2010 | http://bioconductor.org/packages/release/bioc/html/edgeR.html |
| ZEN Microscopy software | Zeiss | https://www.zeiss.com/microscopy/en/products/software/zeiss-zen.html |
| Other | | |

## Experimental model and subject details

Unless stated, *C. elegans* strains were maintained at 20 °C on NGM plates seeded with OP50 bacteria and were handled using standard techniques (Brenner, 1974).

The HSF-1 AID allele, *hsf-1(ljt3[hsf-1::degron::gfp])I*, was from our previous work (Edwards et al, 2021). To perform HSF-1 depletion in strains that express NMY-2::GFP (*zuIs45 [nmy-2p::nmy-2::GFP + unc-119(+)] V*) (Nance et al, 2003) and GFP::H2B (*oxIs279 [pie-1p::GFP::H2B + unc-119(+)]II*) (Frok-jaer-Jensen et al, 2008), we made another HSF-1 AID *allele hsf-1(ljt5[hsf-1::degron::3xFLAG])I* using CRISPR knock-in of degron::3xFLAG to the C-terminus of endogenous *hsf-1* gene through microinjection of chemically modified synthetic sgRNA (Synthego) along with Cas9 Nuclease (Integrated DNA Technologies, IDT) following the previously published protocol (Prior et al, 2017). The repair template was made by NEB assembly of upstream flanking sequence and degron from the repair template of degron::gfp and a synthetic gene fragment from IDT that contains a 3xFLAG tag and the downstream flanking sequence. The new HSF-1 AID model was outcrossed 6 times before use.

The Bristol N2 and other worm strains were obtained from the *Caenorhabditis elegans* Genetics Center (CGC). The alleles used in this study include:

*ieSi38[sun-1p::TIR1::mRuby::sun-1_3′UTR+Cbr-unc-119(+)] IV* (induce AID in the germline)

*daf-2(e1370)III, pept-1(lg601)X, daf-16(mgDf50)I, rsks-1(ok1255) III, rde-1(mkc36)V, eat-6(ad467)V*

*JAR16 rps-6(rns6[rps-6::mCherry])I (measurement of endogenous RPS-6)*

*mkcSi13 [sun-1p::rde-1::sun-1 3′UTR + unc-119(+)]II* (germline RNAi)

*neIs9 [myo-3::HA::rde-1 + rol-6(su1006)]X* (muscle RNAi)

*kzIs9 [(pKK1260) lin-26p::NLS::GFP + (pKK1253) lin-26p::rde-1 + rol-6(su1006)]* (hypodermis RNAi)

*frSi17[mlt-2p:rde-1]II* (intestine RNAi).

## Method details

### HSF-1 depletion and RNAi

HSF-1 depletion was done in the AID models that degraded the HSF-1 protein, specifically in the germline, upon auxin treatment. Auxin treatment was performed by transferring worms to bacteria-seeded NGM plates containing 1 mM auxin (indole-3-acetic acid, Sigma). The preparation of auxin stock solution (400 mM in ethanol) and auxin containing NGM plates was done as previously described (Zhang et al, 2015). In all experiments, worms were also transferred to NGM plates containing 0.25% ethanol (EtOH) as the mock-treated control.

RNAi was performed by feeding, and all RNAi clones from the Ahringer Library (Kamath et al, 2003) were sequence-verified before use. An RNAi-compatible OP50 strain (Xiao et al, 2015) was used in all the experiments. Overnight cultures of RNAi bacteria in LB media containing 100 μg/ml ampicillin were diluted and allowed to grow for another 5–6 h at 37 °C to reach OD600 of 1.0–1.2. Cultures were then seeded onto NGM plates containing 100 μg/ml ampicillin and 1 mM IPTG to induce expression of dsRNA and dry at room temperature for 2 days. All RNAi experiments were done by laying eggs on freshly prepared RNAi plates. Tissue-specific RNAi were performed in transgenic worms expressing the Argonaute protein gene *rde-1* in the germline or specific somatic tissue in the null mutant *rde-1 (mkc36)* (Zou et al, 2019).

### Translation assay

The O-propargyl-puromycin (OPP) translation assay was performed as previously described (Somers et al, 2022), except that fixation and conjugation of fluorescent azide were done in dissected gonads rather than in the whole worms. During incubation with OPP, worms were fed with dead bacteria. OP50 bacteria were killed by incubating with 1% paraformaldehyde for 1 h and washed three times with M9 buffer. In each experiment, 50–60 young adult worms synchronized by egg lay were used. Worms were incubated in 10 μM OPP (Invitrogen) diluted with M9 buffer containing 5 g/L of dead OP50 in a total volume of 1 mL for 3 h at 20 °C with gentle shaking. Animals were then dissected in PBS buffer containing 0.1% Tween-20 (PBST) and 0.25 mM levamisole to expose gonads. The samples were then fixed in PBST buffer containing 3% paraformaldehyde for 10 min at room temperature and then permeabilized with cold methanol for 15 min at −20 °C. After three washes with PBST, the samples were used to conjugate fluorophore to the puromycylated peptides using the Click-iT® Plus Alexa 647 Fluor® Picolyl Azide tool kit (Invitrogen). The pellet of gonads was resuspended and incubated with 200 μL Click-iT® reaction buffer containing 0.5 μL of 500 μM Alexa Fluor® Picolyl Azide, 8 μL 100 mM copper protectant, 20 μL of reaction buffer additive, 17 μL of 10x reaction buffer and 153 μL of pure water for 1 h at 20 °C with gentle shaking. After the Click-iT® reaction, the pellet was washed with reaction rinse buffer (3% bovine serum albumin in PBST) and, subsequently, three times with PBST to remove the unconjugated Alexa Fluor® Picolyl Azide. Finally, DNA staining was done in 50 ng/ml of DAPI for 10 min, and the samples were mounted on an agarose pad for confocal imaging.

### Proteotoxic stress test

The F1 survival assay following acute heat shock (HS) was performed as previously described (Das et al, 2021) with some modifications. Day 1 gravid adults were exposed to a 15-min HS at 34 °C by submerging the culture plates in a water bath and immediately moved back to 20 °C to recover. Eggs laid within 4 h post-HS were scored for embryo lethality.

Proteotoxic stress tests by proteasome inhibitor and paromo-mycin were done on NGM plates seeded with paraformaldehyde-killed OP50. 400 μL M9 buffer containing 10 μM of the proteasome inhibitor, bortezomib (Sigma-Aldrich), or 1 mM of paromomycin (Sigma-Aldrich) was applied to the surface of 3.5 cm seeded NGM plates. Animals were singled to those plates at the young-adult stage

and embryo lethality in the progenies was measured through the reproductive period.

### Fecundity and embryo lethality measurement

As specified in figure legends, animals were synchronized by egg lay on regular NGM plates or plates containing auxin and/or RNAi bacteria. Worms were singled at the young adult stage and allowed egg-laying for 24 h. Worms were transferred to new plates daily, and eggs were allowed to hatch and grow to the L3 stage. The number of viable larvae and dead eggs was counted at this point. In mating experiments, both males and hermaphrodites at the young-adult stage were used. The N2 males were added to hermaphrodites in a 2:1 ratio and kept for 24 h. Only those hermaphrodites that have successful mating (close to 50% male progenies) were included in fecundity and embryo lethality analysis. Amino Acid supplementation was performed as previously described (Spanier et al, 2018). A 1:1 mixture of MEM non-essential amino acids (100×) and MEM amino acids (50×) without L-glutamine (Sigma-Aldrich) was applied to the surface of 3.5 cm NGM plates. The freshly prepared plates were used the next day.

### Germline nuclei isolation

Germline nuclei isolation followed a published protocol (Han et al, 2019) except that worms were lightly fixed in 50 mL of −20 °C dimethylformamide (Sigma-Aldrich) for 2 min and washed three times in PBS before homogenization. This light fixation helped preserve the RNA from degradation. About 25,000 young adult worms from fifty 10 cm NGM plates were used in each nuclei prep. Both the enriched germline nuclei and whole worm lysate were saved for RNA-seq analysis.

### RNA extraction, cDNA synthesis, and qPCR

Animals were synchronized by treatment of alkaline hypochlorite solution or egg laying (for auxin treatment starting from egg lay). When using alkaline hypochlorite, synchronized L1 larvae were grown on 10 cm regular NGM plates (~500 worms per plate) for larval development. When checking DAF-16-dependent *pept-1* expression (Fig. 6), worms were collected at the late L4 stage when *pept-1* expression peaks. For RNA-seq analysis of the *daf-2(e1370)* animals, ~120 adult worms were used in each experiment. The worms were kept on 10 cm NGM plates contain EtOH or auxin since egg lay or picked onto those plates at the young adult stage and kept for 16 h. RNA was extracted using 300 μL Trizol reagent. Worms were vortexed continuously for 20 min at 4 °C and then went through one cycle of freeze-thaw to help release RNA. Following this, RNA was purified using the Direct-zol RNA MiniPrep kit (Zymo Research) per the manufacturer's instructions using on-column DNase I digestion to remove genomic DNA. RNA was used in library preparation for sequencing or to synthesize cDNA for qPCR analysis.

cDNA was synthesized using the BioRad iScript cDNA synthesis kit per the manufacturer's instructions. Relative mRNA levels were then determined by real-time quantitative PCR using iTaq Universal SYBR Green Supermix (BioRad) and a ThermoFisher QuantStudio 6 Pro thermocycler. Relative mRNA levels were calculated using the standard curve method, and gene expression was normalized to the mean of the housekeeping genes *cdc-42* and *rpb-2*. The following primers were used in qPCR:

pept-1-F: ACTATGGAATGAGAACGGT; pept-1-R: CTTGTC CGATTGCGTAT

sod-3-F: CACTGCTTCAAAGCTTGTTCA; sod-3-R: ATGGG AGATCTGGGAGAGTG

rpb-2-F: AACTGGTATTGTGGATCAGGTG; rpb-2-R: TTTGA CCGTGTCGAGATGC

cdc-42-F: TGTCGGTAAAACTTGTCTCCTG; cdc-42-R: ATC CTAATGTGTATGGCTCGC

### RNA-seq analysis

Total RNAs were polyA enriched, and directional RNA-seq libraries were prepared using the NEBNext Ultra II RNA library prep Kit. Paired-end sequencing was done at a NovaSeq 6000 sequencer at the OMRF clinical genomics core.

RNA-seq reads were mapped to Ensembl WBcel235 genome using RNA STAR (Dobin et al, 2013) with --alignIntronMax 120000 to set the intron size, and --outFilterMultimapNmax 200 to allow multi-mapped reads. The mapped reads were then subject to FeatureCounts in Rsubread (Liao et al, 2019) for quantification with the setting -p -B -P -C -M -O --fraction –largestOverlap. The settings in STAR and FeatureCounts enabled proper quantification of those heat shock genes (e.g., *hsp-70* and *hsp-16*s) duplicated in the *C. elegans* genome. Differential expression (DE) analyses were done using edgeR (Robinson et al, 2010) with default settings except for using the Likelihood Ratio Test and filtering out those lowly expressed genes with CPM (counts per million) values less than 1 in more than 75% of samples. We then filtered out DE genes caused by auxin treatment using a list from our previous study (Edwards et al, 2021) and reported only DE genes caused by HSF-1 depletion.

Gene ontology analysis (GO) was done using the program DAVID (http://david.abcc.ncifcrf.gov/) with functional annotation clustering to collapse redundant GO terms. The enrichment score for each cluster was shown with the corresponding GO_BP (Biological Processes) term representing the cluster.

### Peptide uptake assay

Peptide uptake was done as previously described (Meissner et al, 2004). About 30–40 worms at the Late L4/young adult stage were picked into cold M9 buffer and washed twice to remove bacteria. Worm pellets were resuspended in M9 buffer containing 1 mM β-Ala-Lys-AMCA (peptide institute), transferred into a 96-well plate, and incubated at 20 °C for 3 h with gentle shaking. Worms were washed in M9 buffer and mounted onto an agarose pad with levamisole for imaging.

### Immunofluorescence (IF)

IF was performed with dissected gonads as previously described (Kocsisova et al, 2018) with slight modifications. Animals were synchronized by egg lay and grown to the young adult stage. Approximately 30 animals were dissected in PBS buffer containing 0.1% Tween-20 (PBST) and 0.25 mM levamisole. The samples were fixed in a PBST buffer containing 3% paraformaldehyde for 10 min at room temperature and then incubated with cold methanol for 1 hour at −20 °C. After washes with PBST, the samples were blocked for 30 min at room temperature in PBST with 1% bovine serum albumin before the antibodies against histone H3 at 1:800 dilution (rabbit, Abcam) and ubiquitinylated proteins (clone FK2, mouse, Sigma-Aldrich) at 1:200 dilution were added. After overnight incubation at 4 °C and four washes with the blocking solution, Goat anti-Rabbit IgG (H + L) Alexa Fluor Plus 488 (Invitrogen) and anti-mouse-Alexa 647 (Invitrogen) were added to

the samples at 1:1000 dilution, and the samples were incubated for 2 h in the dark at room temperature. DNA staining was done in 100 ng/ml of DAPI for 20 min, and the samples were mounted on a 2% agarose pad for confocal imaging.

### EdU labeling

EdU labeling was performed by feeding animals with EdU-labeled MG1693 bacteria and by click reaction as previously described (Kocsisova et al, 2018). Briefly, the animals were transferred onto M9 plates with EdU bacteria and fed for half an hour. Animals were dissected in PBS buffer containing 0.1% Tween-20 (PBST) and 0.25 mM levamisole, fixed for 10 min in 3% paraformaldehyde (Fisher Scientific) in PBST, washed in PBST, and incubated for one hour in −20 °C methanol. Rehydration was done by washing the samples three times in PBST. EdU click reaction using Click-iT™ EdU Cell Proliferation Kit with Alexa Fluor™ 594 dye (Thermofisher) and DAPI staining were performed as described (Kocsisova et al, 2018). Worms were then mounted on a 2% agarose pad for confocal imaging. Nuclei with any EdU labeling (individual or all chromosomes) were scored positive. Mitotic and transition zones were defined based on the crescent-shaped DAPI staining in the transition zone.

### Acridine Orange (AO) staining

AO staining was performed following the WormBook protocol that is based on previously published reports (Gumienny et al, 1999; Lettre et al, 2004). Briefly, young adult animals that were either HSF-1 depleted from the germline for 24 h or mock-treated were transferred to seeded NGM plates with 500 μl of M9 buffer containing AO (Thermofisher) at 0.02 mg/ml and incubated for 1 h in the dark. Stained worms were then transferred to regular NGM plates lacking AO for 1 h before being mounted on slides for imaging.

### Fluorescence imaging and quantification of fluorescence intensity

Imaging of live animals, as in Figs. 1C, 2C, EV1B–E, 4C, EV5A, and 7A,C, was done by immobilizing animals in a drop of M9 buffer containing 6 mM levamisole on a 2% agarose pad. Images were acquired immediately using a Zeiss LSM880 or LSM980 Confocal Microscope with a 20× or 40× objective. As specified in the figure legend, proteasome inhibitor was applied to a subset of live-imaging experiments. In those experiments, worms were transferred to the M9 buffer containing 10 μM of bortezomib and paraformaldehyde-killed OP50 and incubated in 96-well plates for 6 h at 20 °C with gentle shaking before imaging. Fluorescent imaging for IF, translation assay, and EdU labeling was performed using a Zeiss LSM880 or LSM980 Confocal Microscope through a 40× water objective (IF and translation) or 63× oil objective (EdU). Zen software was used to obtain z-stacks and subsequent analyses. Fluorescence intensity was quantified in individual worms after maximal intensity projection. Regions of interest were outlined within individual worms, and the arithmetic mean of fluorescence intensity per area was determined.

### Biochemical analysis of protein aggregates

The fractionation and analysis of protein aggregates were performed as previously described (Kirstein-Miles et al, 2013). About 3000 age-synchronized adult worms with HSF-1 depletion for 24 h or mock-treated were harvested by washing off the plates using cold M9 buffer and washed twice to remove bacteria. The worms were washed once in ice-cold lysis buffer (20 mM potassium-phosphate, pH 6.8, 1 mM

DTT, 1 mM EDTA, 0.1% (v/v) Tween-20 and protease inhibitor cocktail (Roche) and frozen in liquid nitrogen. The frozen animals were ground with a plastic pestle on dry ice and then resuspended in 1 mL of ice-cold lysis buffer. The worms were further broken up using a glass Dounce tissue grinder. After successful lysis, the lysate was centrifuged for 5 min at 200 × g at 4 °C to remove debris. The supernatant of multiple samples was adjusted to identical protein concentrations using BCA assay (ThermoFisher). Aggregated proteins were isolated by spinning at 18,000 × g for 20 min at 4 °C. The resulting pellet was then sonicated in a Biorupter (Diagenode) using a high energy setting for 6 min with 30 s on/30 s off in washing buffer (2% NP-40, 20 mM potassium-phosphate, pH 6.8, and protease inhibitor cocktail), and subsequently centrifuged at 18,000 × g for 20 min at 4 °C. This washing step was performed twice. Aggregated proteins were then solubilized in 2% SDS sample buffer at 95 °C for 10 min. Proteins were separated by SDS–PAGE and analyzed by immunoblotting. In western blot analysis, primary antibodies against α-tubulin at 1:5000 dilution (mouse, Sigma-Aldrich) and ubiquitin at 1:2000 dilution (mouse, Novus) were used. The blot was then probed by Goat anti-Mouse Poly-HRP secondary antibody at 1:5000 dilution.

## Quantification and statistical analysis

Two-tailed, unpaired Student's t-tests were used for fecundity and embryo lethality comparison, and image quantifications. Two-tailed, paired Student's t-tests were used for RT-qPCR analysis. *P* values from these statistical analyses were calculated using GraphPad Prism, and are declared in figure legends. Error bars represent SEM or standard deviation as specified in figure legends. Benjamini and Hochberg FDR was used to calculate the adjusted *P*-value for differential expression analyses by RNA-seq.

## Data availability

The RNA-seq datasets from this study: Gene Expression Omnibus GSE256186 (https://www.ncbi.nlm.nih.gov/geo/query/acc.cgi?acc=GSE256186).

The RNA-seq analysis of our published data on HSF-1 depletion from the germline of wild-type animals: Gene Expression Omnibus GSE162066 (https://www.ncbi.nlm.nih.gov/geo/query/acc.cgi?acc=GSE162066).

The source data of this paper are collected in the following database record: biostudies:S-SCDT-10_1038-S44318-024-00234-x.

## Peer review information

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

## Acknowledgements

We thank Dr. Jarod Rollins for sharing worm strains, Dr. Shuai Gao for sharing equipment, and Dr. Patricija van Oosten-Hawle for helpful discussion on PQM-1. We thank members of Li Lab for providing feedback on the manuscript. We thank the imaging core facilities at NYMC and OMRF for assistance with confocal imaging and the OMRF clinical genomics core for RNA-seq. This work was supported by the National Institutes of Health grant R35 GM138364 to JL. This work was also supported in part by a shared instrumentation grant for an LSM 980 plus Airyscan 2 for the NYMC Imaging Core 1S10OD028527-01.

## Author contributions

**Tahir Muhammad**: Conceptualization; Resources; Data curation; Formal analysis; Investigation; Methodology; Writing—review and editing. **Stacey L Edwards**: Resources; Data curation; Investigation; Methodology. **Allison C**

**Morphis**: Resources; Data curation; Investigation; Methodology. **Mary V Johnson**: Data curation; Investigation; Methodology; Writing—review and editing. **Vitor De Oliveira**: Investigation; Methodology. **Tomasz Chamera**: Investigation; Methodology. **Siyan Liu**: Investigation. **Ngoc Gia Tuong Nguyen**: Investigation. **Jian Li**: Conceptualization; Resources; Data curation; Formal analysis; Supervision; Funding acquisition; Validation; Investigation; Visualization; Methodology; Writing—original draft; Project administration; Writing—review and editing.

Source data underlying figure panels in this paper may have individual authorship assigned. Where available, figure panel/source data authorship is listed in the following database record: biostudies:S-SCDT-10_1038-S44318-024-00234-x.

## Disclosure and competing interests statement

The authors declare no competing interests.

# Expanded View Figures

**Figure EV1.   Related to Fig. 1. Loss of HSF-1 from the germline impairs the expression of chaperone genes, causing protein degradation and aggregation.**

(**A**) Histograms showing mRNA fold changes (FC) of the *nmy-2* gene upon HSF-1 depletion from the germline of young adults for 8 h, 16 h, or 24 h based on RNA-seq analysis. The chaperone gene *hsp-90* and proteasome component *pas-1* are included as controls. Mean and standard deviation are plotted (8 h & 16 h: $N = 4$; 24 h: $N = 3$). (**B**) Representative images of GFP::H2B transgene in the germline upon HSF-1 depletion. HSF-1 was depleted from the germline of young adults using AID for 24 h in the presence or absence of the proteasome inhibitor bortezomib (PI) for the last 6 h of HSF-1 depletion. The dashed lines outline the gonads. The white asterisks (*) indicate the distal end of gonads where progenitor cells are located. The arrows indicate the fully grown oocytes, where the levels of GFP::H2B are quantified. (**C**) Representative images of the endogenously tagged HSF-1::degron::GFP upon auxin-induced degradation. HSF-1 was depleted from the germline of young adults upon auxin treatment for 24 h in the presence or absence of the proteasome inhibitor bortezomib (PI) for the last 6 h of HSF-1 depletion. The dashed lines outline the gonads. The white asterisks (*) indicate the distal end of gonads where progenitor cells are located. The proteasome inhibitor treatment was not sufficient to restore HSF-1 protein levels. (**D–F**) Representative images of NMY-2::GFP transgene in the germline upon HSF-1 depletion from germ cells starting at the young-adult stage for 24 h (**D**) and 48 h (**E**), and the corresponding quantification (**F**). The dashed lines outline the gonads. The white asterisks (*) indicate the distal end of gonads where progenitor cells are located. In the dot plots, the mean and standard deviation are plotted (Day 2: $N = 8$; Day 3: control $N = 7$, HSF-1 depletion $N = 9$). *P*-values were calculated by unpaired t-test. ns: $p > = 0.05$.

▶

                              *The EMBO Journal* Volume 43 | Issue 21 | November 2024 | 4892 – 4921     **4913**

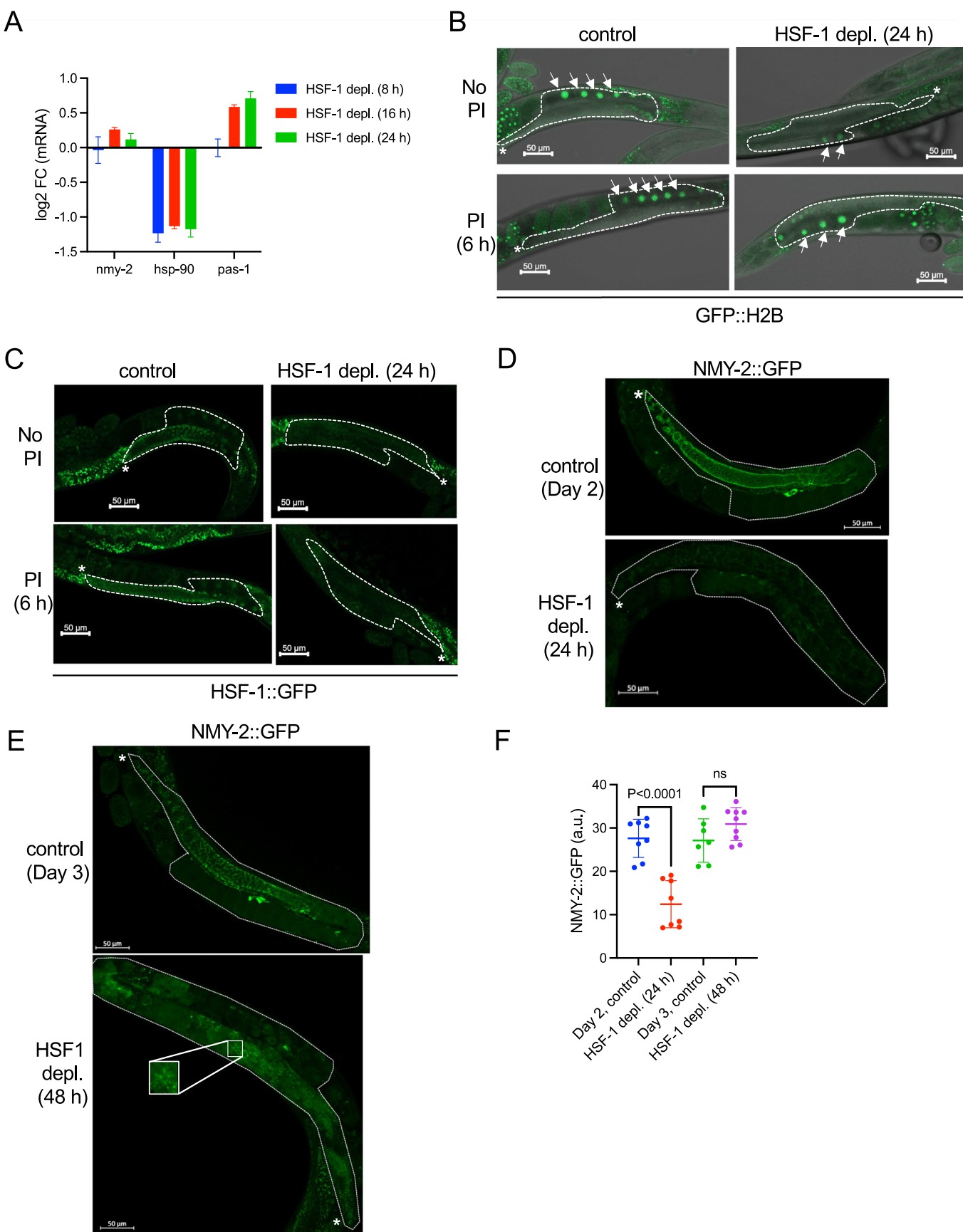

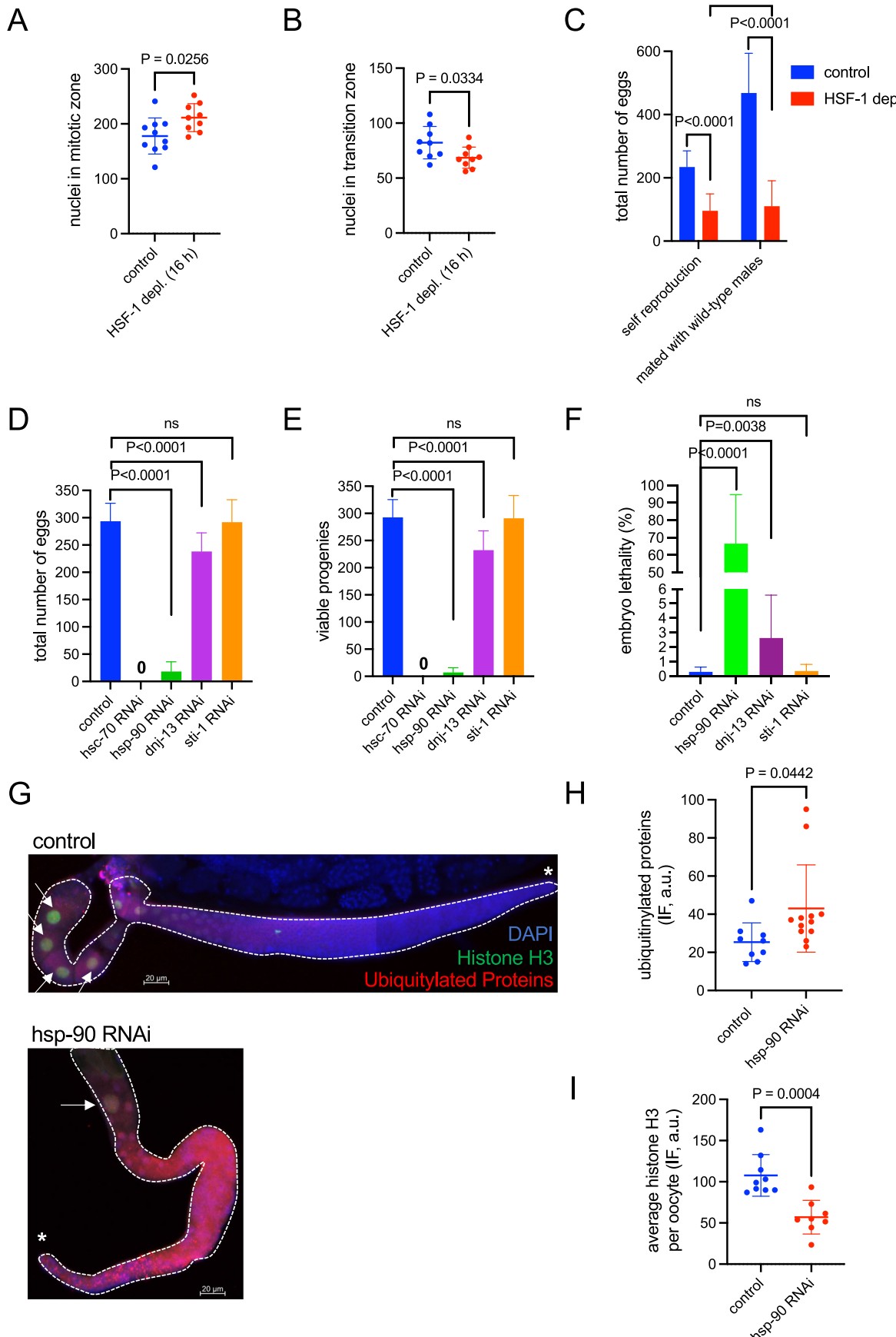

◄ **Figure EV2.   Related to Fig. 2. HSF-1 is required in the adult germline for fecundity and oocyte quality.**

(A, B) Quantification of the total number of nuclei in the mitotic zone (A) and transition zone (B) upon depletion of HSF-1 from the germline of young adults for 16 h. Mean and standard deviation are plotted ($N > = 9$). *P*-values were calculated by unpaired t-test. (C) Histograms showing the total number of eggs (live and dead) upon HSF-1 depletion from the germline of hermaphrodites starting from Day 1 of adulthood during self-reproduction and when mated with N2 males. Mean and standard deviation are plotted (self-reproduction: $N = 15$; mating experiments: $N > = 10$). *P*-values were calculated by unpaired t-test. ns: $p > = 0.05$. (D–F) Histograms showing the total number of eggs (live and dead) (D), viable progenies (E), and embryo lethality (F) with germline-specific RNAi against selective chaperone and co-chaperone genes. The four selected chaperone and co-chaperone genes are direct target genes of HSF-1, which have HSF-1 binding at their promoters in germ cells (ChIP-seq) and significantly decrease mRNA expression upon 8 h of HSF-1 depletion from the germline (FDR: 0.05, RNA-seq) (Edwards et al, 2021). Animals were treated with control RNAi (L4440) or RNAi against chaperone and co-chaperone genes starting from egg lay. Mean and standard deviation are plotted ($N > = 15$). *P*-values were calculated by unpaired t-test. ns: $p > = 0.05$. (G–I) Representative images (G) and quantification of ubiquitylated proteins (not free ubiquitin) (H) and endogenous histone H3 (I) by immunofluorescence (IF) upon germline-specific RNAi against *hsp-90*. Animals were treated with control RNAi (L4440) or RNAi against *hsp-90* starting from the L3 larval stage, which allowed more oocytes to develop in the hsp-90 RNAi group than starting the RNAi treatment at egg lay. IF was performed on Day 1 of adults. The dashed lines outline the gonads with the white asterisks (*) marking the distal end. The arrows indicate the fully grown oocytes, where the levels of histone H3 are quantified. Mean and standard deviation are plotted (control RNAi: $N = 9$; hsp-90 RNAi: $N = 12$ for ubiquitylated proteins and $N = 8$ for histone H3 as not all hsp-90 RNAi treated animals developed fully grown oocytes). *P*-values were calculated by unpaired t-test.

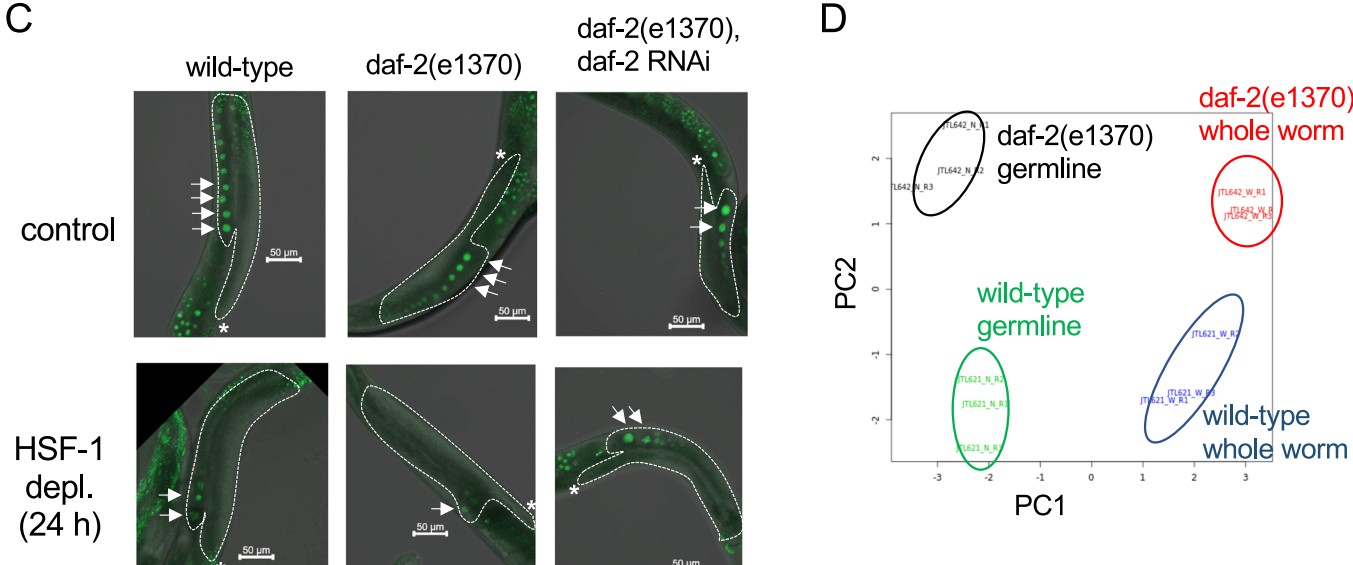

**Figure EV3.  Related to Fig. 3 and Fig. 4. Reduced Insulin/IGF-1 signaling (IIS) confers resilience against limited protein folding capacity in gametogenesis through the regulation of translation.**

(A, B) Representative images (A) of endogenous histone H3 and ubiquitylated proteins (not free ubiquitin) by immunofluorescence (IF) upon germline-specific depletion of HSF-1 in the *daf-2(e1370)* animals starting from the young-adult stage for 24 h. The dashed lines outline the gonads with the white asterisks (*) marking the distal end. The levels of histone H3 in the fully grown oocytes (as indicated by arrows) are quantified (B) together with the results from the wild-type animals as a control. Mean and standard deviation are plotted ($N >= 12$). *P*-values were calculated by unpaired t-test. (C) Representative images of GFP::H2B transgene in fully grown oocytes upon germline-specific depletion of HSF-1 for 24 h from young adults. The wild-type and *daf-2(e1370)* animals were treated with *daf-2* RNAi or control RNAi (L4440) starting from egg lay. The dashed lines outline the gonads with the white asterisks (*) marking the distal end. The arrows indicate the fully grown oocytes, where the levels of GFP::H2B are quantified. (D) Principal component analysis (PCA) on the RNA-seq results of the whole worm lysate or isolated germline nuclei from young adults of the wild-type and *daf-2(e1370)* animals ($N = 3$).

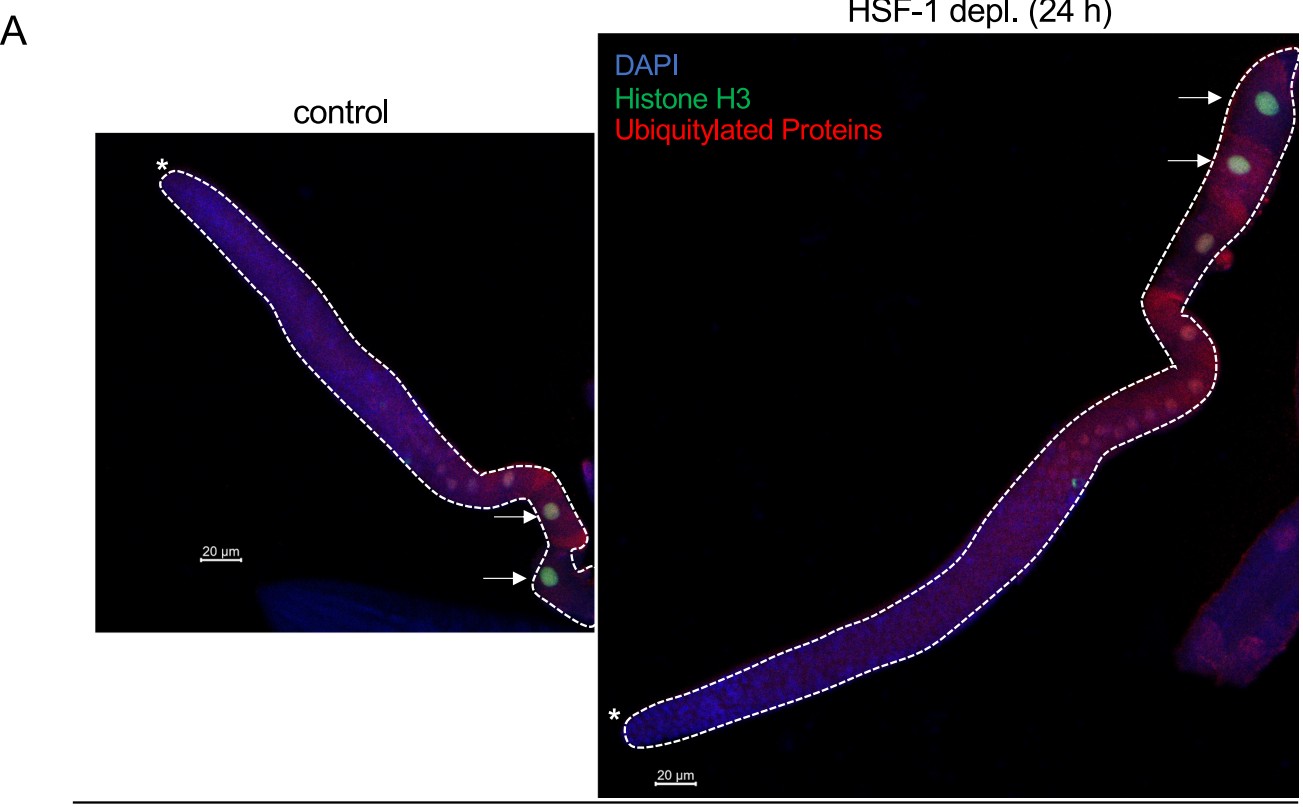

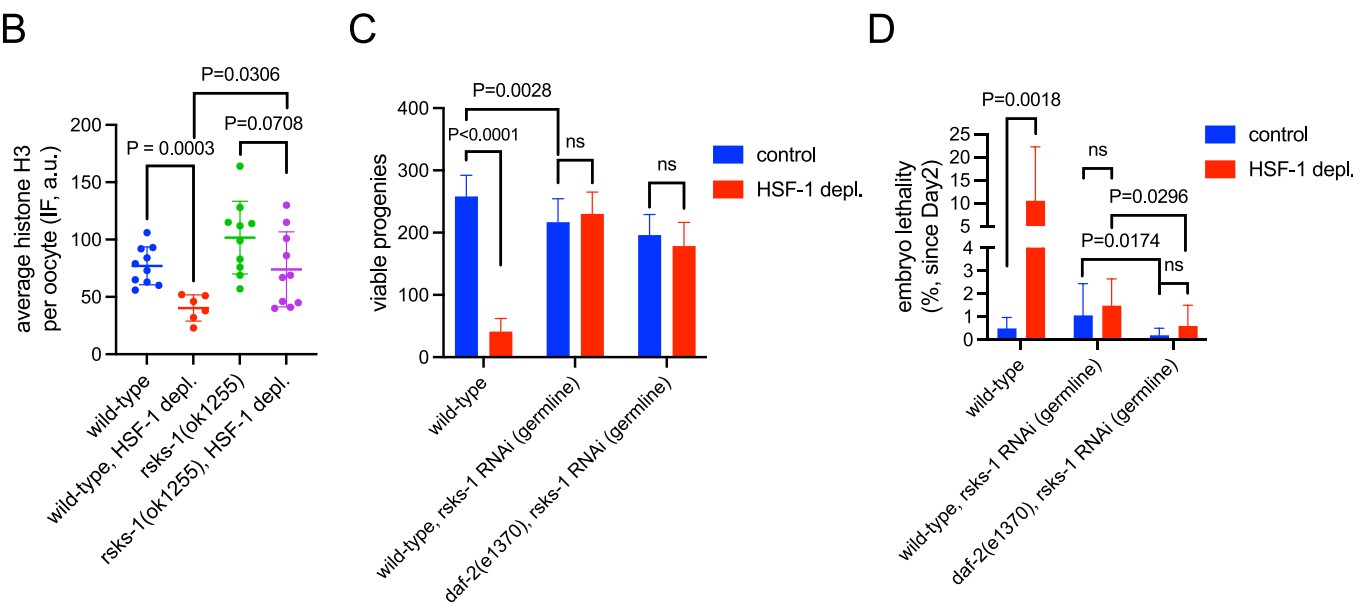

◄ **Figure EV4.    Related to Fig. 5. Reduced translation rate underlies robust germline proteostasis and resilience against proteotoxic stress mediated by low insulin/IGF-1 signaling (IIS).**

(**A, B**) Representative images (**A**) of endogenous histone H3 and ubiquitylated proteins (not free ubiquitin) by immunofluorescence (IF) upon germline-specific depletion of HSF-1 in the *rsks-1(ok1255)* animals starting from the young-adult stage for 24 h. The dashed lines outline the gonads with the white asterisks (*) marking the distal end. The levels of histone H3 in the fully grown oocytes (as indicated by arrows) are quantified (**B**) together with the results from the wild-type animals as a control. The mean and standard deviation are plotted (wild-type, HSF-1 depletion: $N = 6$; the other three groups: $N = 10$). *P*-values were calculated by unpaired t-test. (**C, D**) Histograms showing the brood size (**C**) and embryo lethality (**D**) of the wild-type and *daf-2(1370)* animals treated with germline-specific *rsks-1* RNAi and with depletion of HSF-1 from the germline. The treatment of control RNAi (L4440) or RNAi against *rsks-1* started at the egg lay. HSF-1 depletion from the germline was initiated on Day 1 of adulthood. Embryo lethality was measured on Day 2 and after. Mean and standard deviation are plotted ($N > = 15$). *P*-values were calculated by unpaired t-test. ns: $p > = 0.05$.

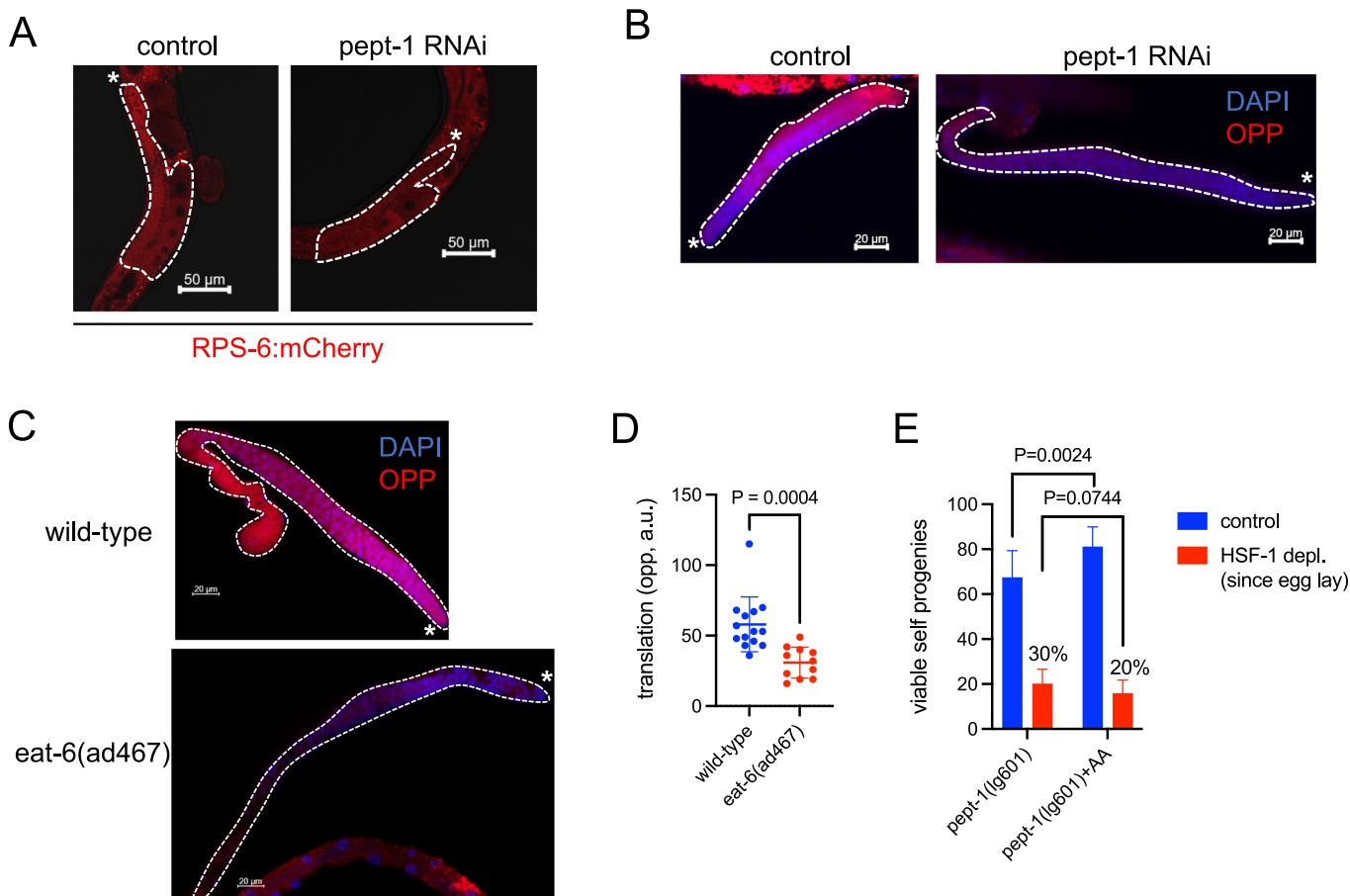

**Figure EV5.** Related to Fig. 7. Insulin/IGF-1 signaling (IIS) regulates germline protein synthesis and proteostasis via peptide uptake in the intestine.

(A, B) Representative images of the endogenously tagged RPS-6::mCherry protein (A) and translation measured by OPP incorporation (B) in the germline upon *pept-1* RNAi. RNAi started at egg lay, and measurement was done in young adults. The dashed lines outline the gonads with the white asterisks (*) marking the distal end. (C, D) Representative images (C) and quantification (D) of translation measured by OPP incorporation in the germline of the wild-type and *eat-6(ad467)* animals. The *eat-6(ad467)* mutant serves as a dietary restriction model. The translation assay was done in young adults. The dashed lines outline the gonads with the white asterisks (*) marking the distal end. Mean and standard deviation are plotted (wild-type: N = 14; *eat-6(ad467)*: N = 11). The P-value was calculated by unpaired t-test. (E) Brood size analysis of the *pept-1(lg601)* animals in the presence or absence of amino acid (AA) supplements. Both depletion of HSF-1 from the germline and the supplement of AA started from egg lay. Mean and standard deviation are plotted (N > = 12). P-values were calculated by unpaired t-test. The percentage of brood size obtained upon HSF-1 depletion compared with the corresponding control is labeled.

