## [Peer Review File · The EMBO Journal]

Non-cell-autonomous regulation of germline proteostasis by insulin/IGF-1 signaling-induced dietary peptide uptake via PEPT-1

Jian Li, Tahir Muhammad, Stacey Edwards, Allison Morphis, Mary Johnson, Vitor Oliveira, Tomasz Chamera, Siyan Liu, and Ngoc Nguyen

Corresponding author: Jian Li (jli37@nymc.edu)

Review Timeline:

Submission Date:	1st Apr 24
Editorial Decision:	13th May 24
Revision Received:	26th Jun 24
Editorial Decision:	30th Jul 24
Revision Received:	8th Aug 24
Accepted:	19th Aug 24

Editor: Kelly Anderson

Transaction Report:

Dear Dr. Li,

Thank you for submitting your manuscript for consideration by the EMBO Journal. It has now been seen by three referees whose comments are shown below.

Given the referees' positive recommendations, I would like to invite you to submit a revised version of the manuscript, addressing the comments of all three reviewers. I should add that it is EMBO Journal policy to allow only a single round of revision, and acceptance of your manuscript will therefore depend on the completeness of your responses in this revised version. It would be good to discuss you plan to address the referee concerns and I am available to do so via zoom or email in the coming weeks.

Thank you for the opportunity to consider your work for publication. I look forward to your revision.

Yours sincerely,

Kelly M Anderson, PhD
Editor, The EMBO Journal
k.anderson@embojournal.org

We realize that it is difficult to revise to a specific deadline. In the interest of protecting the conceptual advance provided by the work, we recommend a revision within 3 months (11th Aug 2024). Please discuss the revision progress ahead of this time with the editor if you require more time to complete the revisions.

Referee #1:

In the manuscript 'Non-cell-autonomous regulation of germline proteostasis by insulin/IGF-2 signaling via the intestinal peptide transporter PEPT-1', the authors show that HSF-1 is required for the regulation of proteostasis during gametogenesis in the *C. elegans* germline. HSF-1 activity has previously been implicated in germ and cancer cell proliferation and oocyte maturation in several organisms. The present work is consistent with and strengthens the contention that HSF1 induces separate transcriptional programs during the heat shock response and development, respectively, and demonstrates the consequences of HSF-1 depletion on fecundity and germ cell quality during nematode development. These results highlight the tissue specificity of the role of HSF-1 in the germline compared to somatic tissues. As previously suggested, HSF-1 activation is induced by the increasing load of protein synthesis associated with gametogenesis. The amount of available data from different model systems that are consistent with the findings presented in this manuscript supports the likelihood that the mechanisms described are conserved during evolution. The authors also show that inhibition of protein synthesis by impairing insulin/IGF-1 signaling (IIS) partially restores gametogenesis in HSF-1-depleted germlines. However, because IIS impairment itself has a dramatic effect on gametogenesis, it is difficult to compare this rescue to wild-type levels of progeny production. The main drawbacks of the study are the lack of use of a more reliable reporter of germline proteostasis in this 'non-stressed' context as well as the lack of investigation of a direct link between nutrient availability and germline proteostasis by including some dietary restriction experiments. The data presented support a model of non-cell-autonomous regulation of germline protein synthesis by IIS in the gut, which adds to the current knowledge of inter-tissue communication in *C. elegans*. The mechanism proposed to underlie this communication involves transcriptional activation of the intestinal peptide transporter PEPT-1, which would highlight a link between nutrient availability and reproduction mediated by IIS and requiring HSF-1 activity. The manuscript is clear and well written and the figures present the necessary information. The Materials and Methods section is well referenced and allows for reproducibility of the experiments performed.

Comments:

1. The efficiency of auxin-induced degradation would be interesting to see either by showing HSF-1::GFP levels in the germline with and without auxin, or by showing a decrease in HSF-1 on a Western blot after auxin treatment.
2. For ease of understanding, it would be helpful to show a diagram of the auxin treatment protocol and the time points at which mRNA and/or protein levels are measured in the different experiments.
3. Figure 2: Are there also fewer cells in the transition zone and in the meiotic pachytene region after HSF-1 depletion? And are there fewer eggs being laid, as would be expected in this situation?
4. Figure 2: Would knockdown of the identified chaperones in the germline have the same effect on embryo lethality and number of viable offspring?
5. Figure 3: The fact that there are fewer ubiquitylated proteins in *daf-2(rf)* could be related to several factors other than less protein misfolding. These include: activation of autophagy for protein clearance in the absence of IIS, less ubiquitylation due to reduced expression of components of the ubiquitin-proteasome system, and lower overall protein levels as a result of reduced translation. Aggregation could be measured here in addition to ubiquitylation to assess the maintenance of proteostasis, or reporters for protein misfolding or folding sensors can be used to monitor misfolding more directly.
6. Another conceptual question would be what is the signaling pathway that promotes increased transcription of UPS components upon HSF-1 depletion, since DAF-16 does not seem to be necessary for this to occur, and how is this inhibited in the absence of DAF-2.
7. Figures 5, 6, 7: Would there be a reduction in function if both *daf-2* and *rsk-1* had an additive effect on improving progeny viability following HSF-1 depletion or would it be the same? Similarly, would *daf-2(rf)* and *pept-1(lf)* have additive effects?

8. Would dietary restriction after HSF-1 depletion have a similar effect on gametogenesis and oocyte integrity as well as ribosome biogenesis and translation rate in the germline as the *daf-2(rf)* and *pept-1(lf)* mutations? This would provide further evidence of how the IIS links nutrient availability to reproduction via HSF-1 activity.

Referee #2:

The authors demonstrate that loss of germline HSF-1 chaperone TF leads to increased levels of ubiquitylated proteins in the germ line. Localize to germline insoluble fraction. The authors show more GSCs when HSF-1 is depleted and impaired GSC proliferation and differentiation. The germ cell proliferation defect is not rescued by mating, so oogenic proliferation is responsible, and high levels of embryonic lethality are a consequence. *Daf-2* deficiency modestly restores germline proliferation but has a pronounced benefit for lethality of embryos from HSF-1 depleted germ cells without restoring expression of chaperone genes. Translation proteins are substantially reduced based on RNA seq, and ribosomal protein levels are shown to decline less in *daf-2* mutants with HSF-1 depleted, as nicely demonstrated by *RPS-1::mCherry* analysis. Remarkably, *rsk-1* mutation suppressed the effects of HSF-1 depletion on fertility, strongly. At a mechanistic levels, this study links expression of *pept-1* amino acid transporter in the intestine to regulation of fertility in germ cells, thereby creating a deeper understanding of how fertility, translation and nutrient availability are coupled to chaperone function in the context of stress.

Comments:

1. The authors show more GSCs when HSF-1 is depleted, but then postulate that '. If this were the case.
2. The authors show that *pept-1* is partially silenced in *daf-2* mutants, which is alleviated in *daf-2; daf-16* mutants. This perhaps in addition to reduced *pept-1* mRNA translation or another form of post-transcriptional repression of *pept-1* may contribute to reduced *pept-1* function, but this is probably not equivalent to the effects of a *pept-1* deletion mutation. This is not to say that deletion of *pept-1* or that overall repression of *pept-1* and associated proteins does not promote resistance to sterility, but it may be inappropriate to suggest causality as the *pept-1* deletion does not mimic the effects of *daf-2*.
3. The authors show that amino acid supplementation of *pept-1* deletion mutants modestly increases fertility for control animals but modestly decreases it when *hsf-1* is depleted. This might indicate that increasing the rate of translation compromises the benefits of *pept-1* depletion. That said, there could be several ways that amino acid supplementation acts and future studies of the effects of *pept-1* on amino acid levels might provide more insight into this hypothesis.
4. The authors point out that they will test a role for autophagy in future work. Given the IIS and S6 kinase mTORC1 results, it is reasonable to suggest that hallmarks of all long lived mutants like autophagy might be involved.

Referee #3:

General Summary

In the manuscript "Non-cell autonomous regulation of germline proteostasis by insulin/IGF-1 signaling via PEPT-1" by Muhammad et al. the authors provide a novel model for the cell non-autonomous control over proteostasis that is an exciting area of study, reported on by several groups, but still poorly understood. Here the authors follow up on their surprising previous findings that protein integrity/proteostasis in the germline is monitored by the core protein quality control machinery regulated by HSF-1, but can be influenced cell non-autonomously by the activity of the insulin signaling pathway in somatic tissue. Their studies detailed in this manuscript support a model whereby the insulin signaling pathway controls the expression of the peptide transporter PEPT-1, and thereby amino acid availability to the germline. This influences protein translation rates, and ultimately the load of proteins that require HSF1 dependent-protein quality control machinery for function to support embryonic viability. The results are novel and interesting, and the model is very elegant. The experiments are for the most part, rigorously and thoroughly conducted. I feel, however, that some important details in the results, and a thorough discussion of some of the results especially where the effect sizes are small, that are missing and should be elaborated on.

Major concerns:

- 1) The authors show alterations in the NMY-2 and H2B in the germline. Is the expression levels of these gene products dependent on HSF1 (directly or indirectly) ? Knowing this could alter the interpretation of the results in Fig 1B. This can be shown by qPCR at the mRNA level, or can be stated with reference to previous studies.
- 2) Figure 1E: The authors should explain better what they conclude from the changes in GFP::H2B. Does this report on oocyte viability, oocyte presence, gametogenesis?
I'm also wondering whether the decrease in intensity of H3 is due to decrease in overall number of germ cells in diakinesis or a decrease in H3 intensity per cell. An inset in Figure 1E (and other images of the gonad) showing a blow-up focusing on oocytes would also be desirable, since DAPI is poorly resolved in the images as presented. Similarly in Fig 2D, the authors show an increase in mitotic cells. It would be important to know whether the defects in oogenesis are because oocytes are not made and cells are accumulating in the mitotic stages of germline proliferation, or whether oocytes are made but degenerate. While this may not change the overall conclusions of the manuscript, it could clarify whether there is a check-point type pausing of oogenesis in the absence of proper protein quality control, or whether there is frank degeneration of oocytes, as is suggested by

the authors in their depiction of protein misfolding.

3) Figure 3D: does downregulating of *dnj-13*, *sti-1*, *hsp-90* and *hsp-70* recapitulate the phenotypes seen in the HSF1 depleted germlines? (e.g. decreased H3, changes in NMY-2::GFP, effects on progeny).

4) Figure 3C: Although the levels of *hsp-90*, *hsc-70*, *dnj-13*, *sti-1* remained depleted in the *daf-2(e1370)* animals, they were higher than in the HSF-1 depleted animals alone. This could suggest that the levels of these chaperones may be what supports oogenesis and progeny viability. This makes the previous critique (#4) important, so as to assess whether the rescue is solely due to alterations in amino acid availability, and translation, as the authors suggest.

5) While the data support a role for PEPT-1 as a modulator of HSF-1 germline phenotype, the effect seen in Fig. 7G is very modest. This suggests that aa availability through PEPT-1 alone may not be sufficiently explain the effects of *daf-2*. The authors should include this in the discussion, and nuance their conclusions accordingly.

Minor concerns:

1) Fig 1A, 3E etc. In all heatmaps, the lanes should be labelled to indicate what they represent. I may have missed this, but I could not find any indication as to what each of the different lanes represent (biological repeats? different time points? etc.)

2) Does dietary restriction also rescue the need for HSF-1 in oocytes?

3) Figure 3A, 5B, etc: Do labels indicate % or number of progeny? (e.g. 300% in control, wild-type ?)

Dear Dr. Anderson,

Thanks for sending us the referee's comments and providing us the opportunity to submit a revised manuscript. We appreciate the referees' constructive suggestions. Please find attached a detailed point-to-point response to the referees' comments and our revised manuscript (including a change-tracked version as a "related manuscript file"). We have performed several key experiments per the referee's suggestions, including those that test the role of insulin/IGF-1 signaling on germline protein aggregation (Fig. 3H), the impacts of dietary restriction on germline proteostasis (Fig. 7G and Fig. EV5 C&D), the effects of RNAi against individual chaperones in the germline (Fig. EV2 D-I), and oocyte apoptosis upon HSF-1 depletion (Fig. 2C&D). We believe the additional data further strengthened the paper. We also addressed the referee's other comments by providing clarification and reference to published work from our lab and colleagues.

Best,

Jian

Referee

#1:

In the manuscript 'Non-cell-autonomous regulation of germline proteostasis by insulin/IGF-2 signaling via the intestinal peptide transporter PEPT-1', the authors show that HSF-1 is required for the regulation of proteostasis during gametogenesis in the *C. elegans* germline. HSF-1 activity has previously been implicated in germ and cancer cell proliferation and oocyte maturation in several organisms. The present work is consistent with and strengthens the contention that HSF1 induces separate transcriptional programs during the heat shock response and development, respectively, and demonstrates the consequences of HSF-1 depletion on fecundity and germ cell quality during nematode development. These results highlight the tissue specificity of the role of HSF-1 in the germline compared to somatic tissues. As previously suggested, HSF-1 activation is induced by the increasing load of protein synthesis associated with gametogenesis. The amount of available data from different model systems that are consistent with the findings presented in this manuscript supports the likelihood that the mechanisms described are conserved during evolution. The authors also show that inhibition of protein synthesis by impairing insulin/IGF-1 signaling (IIS) partially restores gametogenesis in HSF-1-depleted germlines. However, because IIS impairment itself has a dramatic effect on gametogenesis, it is difficult to compare this rescue to wild-type levels of progeny production. The main drawbacks of the study are the lack of use of a more reliable reporter of germline proteostasis in this 'non-stressed' context as well as the lack of investigation of a direct link between nutrient availability and germline

proteostasis by including some dietary restriction experiments. The data presented support a model of non-cell-autonomous regulation of germline protein synthesis by IIS in the gut, which adds to the current knowledge of inter-tissue communication in *C. elegans*. The mechanism proposed to underlie this communication involves transcriptional activation of the intestinal peptide transporter PEPT-1, which would highlight a link between nutrient availability and reproduction mediated by IIS and requiring HSF-1 activity. The manuscript is clear and well written and the figures present the necessary information. The Materials and Methods section is well referenced and allows for reproducibility of the experiments performed.

We appreciate the referee's positive comments and constructive suggestions. In the revised manuscript, we provide new data showing that low insulin/IGF-1 signaling (IIS) reduces protein misfolding/aggregation upon HSF-1 depletion from the germline (Fig.3H), and dietary restriction tunes down protein translation in the germline and partially restores reproduction in the absence of HSF-1 (Fig. 7G and Fig. EV5 C&D).

Comments:

1. The efficiency of auxin-induced degradation would be interesting to see either by showing HSF-1::GFP levels in the germline with and without auxin, or by showing a decrease in HSF-1 on a Western blot after auxin treatment.

We agree with the referee that it is important to control the efficiency and specificity of HSF-1 depletion by auxin-inducible degradation (AID). In our previous work (Edwards et al., 2021), we conducted thorough analyses using live imaging (HSF-1::GFP) and immunofluorescence (using endogenously FLAG-tagged HSF-1) to show that the AID system could deplete HSF-1 specifically in the germline to the background level in both wild-type animals and upon reduction of IIS. We have emphasized this point in the new diagram (Fig. 1B).

2. For ease of understanding, it would be helpful to show a diagram of the auxin treatment protocol and the time points at which mRNA and/or protein levels are measured in the different experiments.

This is a great suggestion. We have added the diagram in Fig. 1B.

3. Figure 2: Are there also fewer cells in the transition zone and in the meiotic pachytene region after HSF-1 depletion? And are there fewer eggs being laid, as would be expected in this situation?

As the referee predicted, the number of nuclei in the transition zone decreases upon HSF-1 depletion (Fig. EV2B), supporting the role of HSF-1 during the transition from mitosis to early meiosis, and consistently, the total number of eggs (hatched + dead) decreases upon HSF-1 depletion (Fig. EV2C).

4. Figure 2: Would knockdown of the identified chaperones in the germline have the same effect on embryo lethality and number of viable offspring?

In our previous work (Edwards et al., 2021), we have shown that germline-specific RNAi of HSF-1-regulated chaperones (HSP-90 and HSC-70) dramatically impaired reproduction. We have reproduced the results and showed that RNAi of the co-chaperone DNJ-13 also reduced brood size and increased embryo lethality, though to a lesser extent (Fig. EV2D-F).

In addition, we showed that RNAi of HSP-90 perturbed the germline proteostasis, as shown by increased levels of ubiquitylated proteins and decreased histone H3 (Fig. EV2G-I).

5. Figure 3: The fact that there are fewer ubiquitylated proteins in *daf-2(rf)* could be related to several factors other than less protein misfolding. These include: activation of autophagy for protein clearance in the absence of IIS, less ubiquitylation due to reduced expression of components of the ubiquitin-proteasome system, and lower overall protein levels as a result of reduced translation. Aggregation could be measured here in addition to ubiquitylation to assess the maintenance of proteostasis, or reporters for protein misfolding or folding sensors can be used to monitor misfolding more directly.

We agree with the referee that the aggregation analysis is an important addition to the ubiquitylation IF to demonstrate that reduced IIS enhances germline proteostasis. We have performed this experiment, and the results indicate that the *daf-2(rf)* mutant dramatically decreased the ubiquitylated protein aggregates upon HSF-1 depletion compared to that in wild-type animals (Fig. 3H). We also included additional discussion (e.g. potential change in autophagy) in the revised manuscript as suggested by the referee.

6. Another conceptual question would be what is the signaling pathway that promotes increased transcription of UPS components upon HSF-1 depletion, since DAF-16 does not seem to be necessary for this to occur, and how is this inhibited in the absence of DAF-2.

We propose that the induction of UPS components is a stress response to misfolded proteins upon HSF-1 depletion. Due to the slower translation rate, the *daf-2(rf)* mutant is more tolerant to limited protein folding capacity and, therefore, results in less protein

misfolding upon HSF-1 depletion (Fig. 3H) and does not trigger the stress response/UPS induction.

SKN-1A/NRF1 is known as the transcriptional activator of UPS components in response to protein misfolding (works from Blackwell lab, Ruvkun lab, and others). Our preliminary data in the SKN-1A null mutant suggest that SKN-1A is required for the expression of the UPS components both in the control (in the presence of HSF-1) and upon HSF-1 depletion. Additional experiments outside the scope of this current study will be needed to dissect the specific signaling pathway that triggers the stress response.

7. Figures 5, 6, 7: Would there be a reduction in function if both *daf-2* and *rsks-1* had an additive effect on improving progeny viability following HSF-1 depletion or would it be the same? Similarly, would *daf-2*(rf) and *pept-1*(lf) have additive effects?

The epistatic analysis suggested by the referee will be very informative on the genetic interactions if loss-of-function (lf) mutants are used. Unfortunately, we must use the reduction-of-function (rf) mutant of *daf-2* in experiments because the loss of IIS completely leads to a larval arrest. Since we have shown that *daf-2* RNAi has additive effects with the *daf-2*(rf) mutant in Figure 3, additive effects of *daf-2* (rf) with *rsks-1* (lf) or *pept-1* (lf) would not determine if they function in the same pathway or parallel pathways.

Nevertheless, we performed germline-specific RNAi against *rsks-1* in both the wild-type and *daf-2*(rf) mutant (Fig. EV4 C&D). Germline-specific *rsks-1* RNAi phenocopied *rsks-1* (lf) in suppressing reproductive defects by HSF-1 depletion, demonstrating the cell-autonomous effect of translation control on germline proteostasis. The *daf-2*(rf) has no additive effects on brood size but small improvements in embryo lethality with *rsks-1* RNAi in the absence of HSF-1. Due to the sickness of the mutant animals and the inconclusive nature of the experiments, we did not perform the assays in the *daf-2*(rf); *pept-1*(lf) double mutant.

8. Would dietary restriction after HSF-1 depletion have a similar effect on gametogenesis and oocyte integrity as well as ribosome biogenesis and translation rate in the germline as the *daf-2*(rf) and *pept-1*(lf) mutations? This would provide further evidence of how the IIS links nutrient availability to reproduction via HSF-1 activity. This is another interesting suggestion. We performed dietary restriction (DR) using a classical *eat-6* mutant that reduces the feeding rate. This DR mutant tunes down protein translation in the germline and partially restores reproduction in the absence of HSF-1 (Figs. 7G and EV5 C&D).

Referee #2:

The authors demonstrate that loss of germline HSF-1 chaperone TF leads to increased levels of ubiquitinated proteins in the germ line. Localize to germline insoluble fraction. The authors show more GSCs when HSF-1 is depleted and impaired GSC proliferation and differentiation. The germ cell proliferation defect is not rescued by mating, so oogenic proliferation is responsible, and high levels of embryonic lethality are a consequence. Daf-2 deficiency modestly restores germline proliferation but has a pronounced benefit for lethality of embryos from HSF-1 depleted germ cells without restoring expression of chaperone genes. Translation proteins are substantially reduced based on RNA seq, and ribosomal protein levels are shown to decline less in daf-2 mutants with HSF-1 depleted, as nicely demonstrated by RPS-1::mCherry analysis. Remarkably, rsk-1 mutation suppressed the effects of HSF-1 depletion on fertility, strongly. At a mechanistic levels, this study links expression of pept-1 amino acid transporter in the intestine to regulation of fertility in germ cells, thereby creating a deeper understanding of how fertility, translation and nutrient availability are coupled to chaperone function in the context of stress.

Comments:

1. The authors show more GSCs when HSF-1 is depleted, but then postulate that ' . If this were the case.

Upon HSF-1 depletion from the young adult germline, we observed a smaller fraction of germ cells in the mitotic zone are in the S-phase (Fig. 2B), implicating impaired proliferation. This is consistent with our previous EdU labeling results when depleting HSF-1 in the germline during larval development and our RNA-seq data that indicate genes involved in DNA replication and cell division are downregulated (Edwards et al 2021). We added this statement and referred to our previous work in the revised manuscript to clarify (Page 6). In the meanwhile, we saw an increased number of nuclei in the mitotic zone (Fig. EV2A) and decreased nuclei number in the transition zone (newly added Fig. EV2B). Given that cell proliferation decreased, we believe the transition of progenitor cells into meiosis (differentiation) is also impaired so that cells accumulate in the mitotic zone.

2. The authors show that pept-1 is partially silenced in daf-2 mutants, which is alleviated in daf-2; daf-16 mutants. This perhaps in addition to reduced pept-1 mRNA translation or another form of post-transcriptional repression of pept-1 may contribute to reduced pept-1 function, but this is probably not equivalent to the effects of a pept-1 deletion mutation. This is not to say that deletion of pept-1 or that overall repression of pept-1 and associated proteins does not promote resistance to sterility, but it may be inappropriate to suggest causality as the pept-1 deletion does not mimic the effects of daf-2.

We appreciate this helpful comment and agree with the referee that pept-1 contributes to the regulation of germline proteostasis by IIS but is probably not the only regulator that links somatic IIS signals to the germline. We have found that the activity of DAF-16 in the muscle also contributes to the regulation of germline sensitivity to HSF-1 depletion; therefore, additional factors in the muscle are likely involved (discussion on Page 15, L3-5). We have softened the conclusion by changing the subtitle and corresponding discussion in the last session of discussion (Page 15, L8-12).

3. The authors show that amino acid supplementation of pept-1 deletion mutants modestly increases fertility for control animals but modestly decreases it when hsf-1 is depleted. This might indicate that increasing the rate of translation compromises the benefits of pept-1 depletion. That said, there could be several ways that amino acid supplementation acts and future studies of the effects of pept-1 on amino acid levels might provide more insight into this hypothesis.

This is another helpful comment. We cannot rule out the possibility that AA supplementation could have additional effects than changing the translation rate, especially since the literature has suggested the levels of different amino acids in the pept-1 mutant are not affected equally. We have included this point in the discussion (Page 15, last sentence).

4. The authors point out that they will test a role for autophagy in future work. Given the IIS and S6 kinase mTORC1 results, it is reasonable to suggest that hallmarks of all long lived mutants like autophagy might be involved.

We agree with the referee that given the broad role of IIS and mTORC1, many branches of the proteostasis network (PN), from synthesis and transport to degradation, could contribute to germline proteostasis. We have expanded our discussion from autophagy to the more general PN players (Page 16, last sentence of the 2nd paragraph).

Referee #3:

General Summary

In the manuscript "Non-cell autonomous regulation of germline proteostasis by insulin/IGF-1 signaling via PEPT-1" by Muhammad et al. the authors provide a novel model for the cell non-autonomous control over proteostasis that is an exciting area of study, reported on by several groups, but still poorly understood. Here the authors follow up on their surprising previous findings that protein integrity/proteostasis in the germline is monitored by the core protein quality control machinery regulated by HSF-1, but can be influenced cell non-autonomously by the activity of the insulin signaling pathway in

somatic tissue. Their studies detailed in this manuscript support a model whereby the insulin signaling pathway controls the expression of the peptide transporter PEPT-1, and thereby amino acid availability to the germline. This influences protein translation rates, and ultimately the load of proteins that require HSF1 dependent-protein quality control machinery for function to support embryonic viability.

The results are novel and interesting, and the model is very elegant. The experiments are for the most part, rigorously and thoroughly conducted. I feel, however, that some important details in the results, and a thorough discussion of some of the results especially where the effect sizes are small, that are missing and should be elaborated on.

Major concerns:

1) The authors show alterations in the NMY-2 and H2B in the germline. Is the expression levels of these gene products dependent on HSF1 (directly or indirectly) ? Knowing this could alter the interpretation of the results in Fig 1B. This can to be shown by qPCR at the mRNA level, or can be stated with reference to previous studies.

We appreciate the referee's comment. We have included the RNA-seq data of NMY-2 showing its expression does not decrease upon HSF-1 depletion (Fig. EV1A). Therefore, the decline of NMY-2::GFP fusion protein (expressed by the NMY-2 promoter) is not due to decreased transcription. Upon HSF-1 depletion, the number of oocytes decreases, where H2B accumulates. Therefore, our whole animal RNA-seq data would not provide an accurate measure of H2B expression change in the germline. However, our proteasome inhibition data (Fig. 1C-E) suggest at least partially the change of NMY-2 and H2B fusion proteins is due to proteasome degradation.

2) Figure 1E: The authors should explain better what they conclude from the changes in GFP::H2B. Does this report on oocyte viability, oocyte presence, gametogenesis?

I'm also wondering whether the decrease in intensity of H3 is due to decrease in overall number of germ cells in diakinesis or a decrease in H3 intensity per cell.

We quantify our GFP::H2B and H3 IF data as the average signal per oocyte in each gonad arm; therefore, the decreased signal indicates a decline in protein levels upon HSF-1 depletion but not a decrease in oocyte viability or presence. As proteasome inhibitor treatment could partially rescue the protein levels, it suggests GFP::H2B and endogenous H3 become unstable and prone to proteasome degradation upon HSF-1 depletion.

An inset in Figure 1E (and other images of the gonad) showing a blow-up focusing on oocytes would also be desirable, since DAPI is poorly resolved in the images as presented.

In all IF experiments, images were obtained by a 40x water objective. DAPI staining only provides staining of chromatin to locate the germline nuclei and does not provide structure details of bivalents in oocytes. The DAPI images in our published work (Edwards et al., 2021) and the EdU labeling in Fig 2A were obtained by a 63x oil objective, which showed the meiotic arrest based on chromatin structure and allowed the distinguishment of mitotic vs. transition zone nuclei.

Similarly in Fig 2D, the authors show an increase in mitotic cells. It would be important to know whether the defects in oogenesis are because oocytes are not made and cells are accumulating in the mitotic stages of germline proliferation, or whether oocytes are made but degenerate. While this may not change the overall conclusions of the manuscript, it could clarify whether there is a check-point type pausing of oogenesis in the absence of proper protein quality control, or whether there is frank degeneration of oocytes, as is suggested by the authors in their depiction of protein misfolding.

Mitosis and meiosis defects certainly contribute to the decrease in oocyte production. To address the question of degeneration (an excellent suggestion), we have included new data showing that oocyte apoptosis increases upon HSF-1 depletion (Fig. 2C&D).

3) Figure 3D: does downregulating of dnj-13, sti-1, hsp-90 and hsp-70 recapitulate the phenotypes seen in the HSF1 depleted germlines? (e.g. decreased H3, changes in NMY-2::GFP, effects on progeny).

We have performed germline-specific RNAi against these chaperone and co-chaperone genes (Fig EV2 D-I). RNAi against hsp-90, hsc-70, and dnj-13 leads to defects in reproduction and oocyte quality, in which hsp-90 and hsc-70 knockdown caused severe defects as upon HSF-1 depletion. We also chose hsp-90 RNAi to test the levels of H3 and protein ubiquitylation (hsc-70 RNAi leads to no oocytes to quantify histone levels) and showed it phenocopied the defects in proteostasis upon HSF-1 depletion.

4) Figure 3C: Although the levels of hsp-90, hsc-70, dnj-13, sti-1 remained depleted in the daf-2(e1370) animals, they were higher than in the HSF-1 depleted animals alone. This could suggest that the levels of these chaperones may be what supports oogenesis and progeny viability. This makes the previous critique (#4) important, so as to assess whether the rescue is solely due to alterations in amino acid availability, and translation, as the authors suggest.

RNA-seq data in Fig.3C show the fold change of chaperone expression upon HSF-1 depletion but do not provide a direct comparison of residual chaperone transcripts in wild-type vs. daf-2(rf) in the absence of HSF-1. As in the presence of HSF-1, daf-2 (rf) has lower chaperone expression, this may give the referee the impression that the residual chaperone levels in daf-2(rf) are higher (as fold change is smaller). However,

when we directly compared the residual levels of HSP-90 and HSC-70 in the wild-type and *daf-2(rf)* by RNA FISH upon HSF-1 depletion, they were the same (Edwards et al 2021). We have added this important point to the revised manuscript. We appreciate the referee pointing out the confusion and have revised the manuscript to make this point clear (Page 7, the 2nd paragraph).

The referee also correctly states that the residual chaperones support oogenesis in the *daf-2(rf)* mutant upon HSF-1 depletion. Our previous work (Edwards et al 2021) has shown that HSF-1 tunes up the chaperone expression rather than serving as an on/off switch. We have shown that the HSF-1-independent basal expression of chaperones is important for the fertility of *daf-2(rf)* animals upon HSF-1 depletion using RNAi against HSP-90 and HSC-70. We want to emphasize in this manuscript that wild-type animals (high IIS) are more addicted to HSF-1-dependent chaperone expression due to their higher rate of translation and requirement for higher protein folding capacity.

5) While the data support a role for PEPT-1 as a modulator of HSF-1 germline phenotype, the effect seen in Fig. 7G is very modest. This suggests that aa availability through PEPT-1 alone may not be sufficiently explain the effects of *daf-2*. The authors should include this in the discussion, and nuance their conclusions accordingly. We agree with the referee that the effect of amino acid supplementation in Fig. 7G (7H in the revised manuscript) is relatively small. This is likely because PEPT-1 mediated peptide uptake is the major pathway of protein absorption, while free amino acid (AA) uptake by separate transporters is the minor pathway. The AA supplement may already exceed the capacity of AA transporters and thus cannot completely rescue the defects of the PEPT-1 mutant. The level of rescue in fecundity (~20%) in Fig. 7H is consistent with the literature. Nevertheless, if we calculate the ratio of brood size of *pept-1(lg601)* upon HSF-1 depletion vs. control (without depletion), AA supplement almost decreased it by half (from 39% to 21%), which suggests significantly increased sensitivity to HSF-1 depletion by AA supplementation. We have added the ratios to Fig. 7H and Fig. EV.5E.

We also agree with the referee that *daf-2* could impact germline proteostasis via additional pathways as *daf-16* activity in the muscle is involved (Fig. 6A). We have revised the discussion in the manuscript to make those arguments (Page 15, L3-L4, L8-12, and the last three sentences).

Minor concerns:

1) Fig 1A, 3E etc. In all heatmaps, the lanes should be labelled to indicate what they represent. I may have missed this, but I could not find any indication as to what each of the different lanes represent (biological repeats? different time points? etc.)

The lanes in heat maps represent biological repeats. We have added this clarification in the figure legends.

2) Does dietary restriction also rescue the need for HSF-1 in oocytes?

We performed dietary restriction (DR) using a classical eat-6 mutant that reduces the feeding rate. This DR mutant tunes down protein translation in the germline and partially restores reproduction in the absence of HSF-1 (Figs. 7G and EV5 C&D).

3) Figure 3A, 5B, etc: Do labels indicate % or number of progeny? (e.g. 300% in control, wild-type ?)

The y-axis label indicates the number of progenies, and the additional % labeling in the figures indicates the percent of progeny compared to the control without HSF-1 depletion in the same genetic background. We have added this clarification to the figure legends.

Dear Dr. Li,

Congratulations on a great revision! Overall, the referees have been positive however one referee had two remaining concerns that we ask you to (non-experimentally) address in a revised version. When you submit your revised version, please also take care of the following editorial items and add this also to your point-by-point response:

1. Please upload the main and EV figures as individual, high resolution figure files.
2. Please include up to five keywords, which may or may not appear in the title, should be given in alphabetical order, below the abstract, each separated by a slash (/).
3. We received error messages for the following email addresses, please correct: tomasz-chamera@omrf.org and Allison C. Morphis - allison-morphis@omrf.org
4. Please update the format of the Data Availability section according to the author guidelines and move this to the end of the Methods.
5. Also in the data availability section, please remove the statement regarding requests for source data and other requests.
6. Please provide the specific IRLs for GSE256186 and GSE162066 datasets in the data availability section.
7. Please merge the funding information with acknowledgements.
8. Please remove the author contribution section from the manuscript.
9. Please rename the conflict of interests statement to Disclosure and Competing Interests.
10. Please include the source data checklist along with the provided source data.
11. We do not allow references to data not shown. Please remove this from page 14.
12. We include a synopsis of the paper (see <http://emboj.embopress.org/>). Please provide me with a general summary statement and 3-5 bullet points that capture the key findings of the paper.
13. We also need a summary figure for the synopsis. The size should be 550 wide by 200-440 high (pixels). You can also use something from the figures if that is easier.
14. We require that all figures be referenced in the main manuscript. Please include a figure call out for Fig 3H in the main manuscript.
15. Please correct the EV Figure legends to: Expanded View Figure Legends. And the figures to "Figure EV1", etc.
16. Please define the annotated p values * as well as provide the exact p-values for the same in the legend of figure EV 2h-i; as appropriate.
17. Please note that the exact p values are not provided in the legends of figures 1d-e, g-h; 2b, d, g; 3a-b, f-g; 4d, f; 5a-e; 6a, c-e; 7b, d-f, h; EV 1f; EV 2a-f; EV 3b; EV 4b-d; EV 5d-e.
18. Please indicate the statistical test used for data analysis in the legends of figures EV 2h-i.
19. Please note that in figure 2b; there is a mismatch between the annotated p values in the figure legend and the annotated p values in the figure file that should be corrected.
20. Please note that information related to n is missing in the legends of figures 1d-e, g-h; 2b, d; 3f-g; 4d, f; 5a; 6d-e; EV 1f; EV 2h-i; EV 3b; EV 4b; EV 5d.
21. Please note that the error bars are not defined in the legends of figures EV 2h-i.
22. Please note that the asterisk is not defined in the legend of figure EV 5a-c. This needs to be rectified.

Thank you for the opportunity to consider your work for publication, I look forward to your revision.

Warm wishes,
Kelly

Yours sincerely,

Kelly M Anderson, PhD
Editor, The EMBO Journal
k.anderson@embojournal.org

Referee #1:

The Authors addressed all my comments and questions, as well as those of the other reviewers.

Referee #2:

The paper by Muhammed, Li and colleagues has been nicely revised, with a number of experiments requested by the reviewers. The paper nicely demonstrates that HSF-1 loss reduces fertility and dramatically increases aggregated and ubiquitylated proteins. Fertility is compromised by reduced differentiation into meiotic germ cells and by increased apoptosis of mature oocytes. The authors demonstrate dramatic reduction of embryo lethality when daf-2 function is strongly reduced and that protein aggregation upon HSF-1 depletion is also dramatically reduced by daf-2 mutation. The authors show that mutation of the gene *pept-1* whose expression is reduced by daf-2 mutation limited peptide intake, similar to a daf-2 mutant, and also rescues fertility. Overall, this well written manuscript presents a satisfying series of experiments that demonstrate how loss of heat shock factor perturbs fertility and how anti-aging pathways restore fertility in this context.

minor points -

1. 'activation of protein synthesis underlies IIS's role in promoting gametogenesis'. The authors show reduced translation, do they mean reduced levels of protein synthesis? Or perhaps clarify that high IIS activity in wild type activates protein synthesis.
2. Similarly in the Discussion 'we found that IIS activates germline protein synthesis'. Modify 'we found that wild type IIS activates'

Referee #3:

In the manuscript 'Non-cell-autonomous regulation of germline proteostasis by insulin/IGF-2 signaling via the intestinal peptide transporter PEPT-1', the authors show that the insulin signaling pathway controls the expression of the peptide transporter PEPT-1, and thereby amino acid availability to the germline. The manuscript supports previous studies that suggest that HSF1 induces separate transcriptional programs during the heat shock response and development, but goes further to provide an elegant framework to understand this difference in regulatory function. HSF-1 activity has previously been implicated in germ and cancer cell proliferation and oocyte maturation in several organisms. Therefore this study has important implications beyond *C. elegans*. In this revised manuscript the authors have convincingly and thoroughly addressed all my concerns, and have much strengthened the manuscript.

I have no further concerns and deem this work to be well suited for publication.

Congratulations on a great revision! Overall, the referees have been positive. However, one referee had two remaining concerns that we ask you to (non-experimentally) address in a revised version. When you submit your revised version, please also take care of the following editorial items and add this also to your point-by-point response:

We appreciate the positive comments from all the referees. We have taken Referee #2's suggestions and changed the wording of our statement about IIS's impact on protein synthesis to avoid confusion. As detailed below, we have also addressed the listed editorial items. Particularly, we have added the exact P-values to respective figures (Item #17).

Referee #2:

minor points -

1. 'activation of protein synthesis underlies IIS's role in promoting gametogenesis'. The authors show reduced translation, do they mean reduced levels of protein synthesis? Or perhaps clarify that high IIS activity in wild type activates protein synthesis.

As the reviewer suggested, we changed the wording on Page 8 to "our results strongly suggest that high IIS activity in wild-type animals may promote gametogenesis via the regulation of protein synthesis".

2. Similarly in the Discussion 'we found that IIS activates germline protein synthesis'. Modify 'we found that wild type IIS activates'

As the reviewer suggested, we changed the wording on Page 13 to "high IIS activity, as in wild-type animals, promotes germline protein synthesis".

1. Please upload the main and EV figures as individual, high-resolution figure files.

We have uploaded the main and EV figures as individual files.

2. Please include up to five keywords, which may or may not appear in the title, should be given in alphabetical order, below the abstract, each separated by a slash (/).

We have included the keywords in the revised manuscript.

3. We received error messages for the following email addresses, please correct: tomasz-chamera@omrf.org and Allison C. Morphis - allison-morphis@omrf.org

Both authors have moved. Their current email addresses are tomasz.chamera@case.edu (Tomasz Chamera) and aclaybrook4@gmail.com (Allison Morphis).

4. Please update the format of the Data Availability section according to the author's guidelines and move this to the end of the Methods.

We have moved the Data Availability section to the end of the Methods.

5. Also in the data availability section, please remove the statement regarding requests for source data and other requests.

We have removed the statement on requests for source data and resources from Data Availability.

6. Please provide the specific IRLs for GSE256186 and GSE162066 datasets in the data availability section.

We included the URLs for both GSE datasets in the Data Availability section.

7. Please merge the funding information with acknowledgments.

We merged the funding information with Acknowledgements.

8. Please remove the author contribution section from the manuscript.

We have removed the author contribution section.

9. Please rename the conflict of interests statement to Disclosure and Competing Interests.

We have renamed the Disclosure and Competing Interests section.

10. Please include the source data checklist along with the provided source data.

We have included the source data checklist and provided comments on the panel number change and the new panels added to the revised manuscript.

11. We do not allow references to data not shown. Please remove this from page 14.

We have removed the discussion on the data not shown on Page 14.

12. We include a synopsis of the paper. Please provide me with a general summary statement and 3-5 bullet points that capture the key findings of the paper.

Please find the summary statement and bullet points below.

Germline proteostasis is essential for gametogenesis, but how it is regulated at the organismal level is poorly understood. Using *C. elegans* as a model, this study shows that germline proteostasis relies on coordinated activities of HSF-1, which functions in germ cells to dictate protein folding capacity, and insulin/IGF-1 signaling (IIS), which controls germline protein synthesis cell-non-autonomously via regulation of dietary protein uptake in the intestine.

-HSF-1 is required from the germline at ambient temperature for chaperone expression, protein folding, and oogenesis.

-Reduced IIS enhances germline proteostasis upon HSF-1 depletion and proteotoxic stress by lowering ribosome biogenesis and translation rate.

-IIS activates the expression of the intestinal peptide transporter PEPT-1, providing a mechanism for regulating germline protein synthesis via dietary protein uptake.

13. We also need a summary figure for the synopsis. The size should be 550 wide by 200-440 high (pixels). You can also use something from the figures if that is easier.

We have included a summary figure "model_pept1.pdf" (539x425, based on Fig. 7I).

14. We require that all figures be referenced in the main manuscript. Please include a figure call out for Fig 3H in the main manuscript.

We apologize for the error. The discussion on Fig. 3H is now added to page 8.

15. Please correct the EV Figure legends to: Expanded View Figure Legends. And the figures to "Figure EV1", etc.

We have changed EV Figure legends as suggested.

16. Please define the annotated p values * as well as provide the exact p-values for the same in the legend of figure EV 2h-i; as appropriate.

We have added the exact p-values to the figure and explained the statistical test in the legend.

17. Please note that the exact p values are not provided in the legends of figures 1d-e, g-h; 2b, d, g; 3a-b, f-g; 4d, f; 5a-e; 6a, c-e; 7b, d-f, h; EV 1f; EV 2a-f; EV 3b; EV 4b-d; EV 5d-e.

We have added the exact p-values to the figures.

18. Please indicate the statistical test used for data analysis in the legends of figures EV 2h-i.

We have added the exact p-values to the figure and explained the statistical test in the legend.

19. Please note that in figure 2b; there is a mismatch between the annotated p values in the figure legend and the annotated p values in the figure file that should be corrected.

We have corrected the legend for Fig. 2B.

20. Please note that information related to n is missing in the legends of figures 1d-e, g-h; 2b, d; 3f-g; 4d, f; 5a; 6d-e; EV 1f; EV 2h-i; EV 3b; EV 4b; EV 5d.

We have added the information related to N to the figure legends.

21. Please note that the error bars are not defined in the legends of figures EV 2h-i.

We have added the information on the error bars to the legend.

22. Please note that the asterisk is not defined in the legend of figure EV 5a-c. This needs to be rectified.

We have added the information on the asterisk to the legend.

Dear Dr. Li,

Congratulations on an excellent manuscript, I am pleased to inform you that your manuscript has been accepted for publication in The EMBO Journal. Thank you for your comprehensive response to the referee concerns and for providing detailed source data. It has been a pleasure to work with you to get this to the acceptance stage.

I will begin the final checks on your manuscript before submitting to the publisher next week. Once at the publisher, it will take about 3 weeks for your manuscript to be published online. As a reminder, the entire review process, including referee concerns and your point-by-point response, will be available to readers.

I will be in touch throughout the final editorial process until publication. In the meantime, I hope you find time to celebrate!

Warm wishes,
Kelly

Kelly M Anderson, PhD
Editor, The EMBO Journal
k.anderson@embojournal.org
